# LASSO with Non-linear Measurements is Equivalent to One With Linear Measurements

**Christos Thrampoulidis,**
Department of Electrical Engineering
Caltech
cthrampo@caltech.edu

**Ehsan Abbasi**
Department of Electrical Engineering
Caltech
eabbasi@caltech.edu

**Babak Hassibi**
Department of Electrical Engineering
Caltech
hassibi@caltech.edu *

## Abstract

Consider estimating an unknown, but structured (e.g. sparse, low-rank, etc.), signal $\mathbf{x}_0 \in \mathbb{R}^n$ from a vector $\mathbf{y} \in \mathbb{R}^m$ of measurements of the form $y_i = g_i(\mathbf{a}_i^T \mathbf{x}_0)$, where the $\mathbf{a}_i$'s are the rows of a known measurement matrix $\mathbf{A}$, and, $g(\cdot)$ is a (potentially unknown) nonlinear and random link-function. Such measurement functions could arise in applications where the measurement device has nonlinearities and uncertainties. It could also arise by design, e.g., $g_i(x) = \text{sign}(x + z_i)$, corresponds to noisy 1-bit quantized measurements. Motivated by the classical work of Brillinger, and more recent work of Plan and Vershynin, we estimate $\mathbf{x}_0$ via solving the Generalized-LASSO, i.e., $\hat{\mathbf{x}} := \arg\min_{\mathbf{x}} \|\mathbf{y} - \mathbf{A}\mathbf{x}_0\|_2 + \lambda f(\mathbf{x})$ for some regularization parameter $\lambda > 0$ and some (typically non-smooth) convex regularizer $f(\cdot)$ that promotes the structure of $\mathbf{x}_0$, e.g. $\ell_1$-norm, nuclear-norm, etc. While this approach seems to naively ignore the nonlinear function $g(\cdot)$, both Brillinger (in the non-constrained case) and Plan and Vershynin have shown that, when the entries of $\mathbf{A}$ are iid standard normal, this is a good estimator of $\mathbf{x}_0$ up to a constant of proportionality $\mu$, which only depends on $g(\cdot)$. In this work, we considerably strengthen these results by obtaining explicit expressions for $\|\hat{\mathbf{x}} - \mu\mathbf{x}_0\|_2$, for the *regularized* Generalized-LASSO, that are asymptotically *precise* when $m$ and $n$ grow large. A main result is that the estimation performance of the Generalized LASSO with non-linear measurements is *asymptotically the same* as one whose measurements are linear $y_i = \mu\mathbf{a}_i^T\mathbf{x}_0 + \sigma z_i$, with $\mu = \mathbb{E}\gamma g(\gamma)$ and $\sigma^2 = \mathbb{E}(g(\gamma) - \mu\gamma)^2$, and, $\gamma$ standard normal. To the best of our knowledge, the derived expressions on the estimation performance are the first-known precise results in this context. One interesting consequence of our result is that the optimal quantizer of the measurements that minimizes the estimation error of the Generalized LASSO is the celebrated Lloyd-Max quantizer.

## 1 Introduction

**Non-linear Measurements.** Consider the problem of estimating an unknown signal vector $\mathbf{x}_0 \in \mathbb{R}^n$ from a vector $\mathbf{y} = (y_1, y_2, \ldots, y_m)^T$ of $m$ measurements taking the following form:

$$y_i = g_i(\mathbf{a}_i^T \mathbf{x}_0), \quad i = 1, 2, \ldots, m. \tag{1}$$

Here, each $\mathbf{a}_i$ represents a (known) measurement vector. The $g_i$'s are independent copies of a (generically random) link function $g$. For instance, $g_i(x) = x + z_i$, with say $z_i$ being normally

distributed, recovers the standard linear regression setup with gaussian noise. In this paper, we are particularly interested in scenarios where $g$ is *non-linear*. Notable examples include $g(x) = \text{sign}(x)$ (or $g_i(x) = \text{sign}(x + z_i)$) and $g(x) = (x)_+$, corresponding to 1-bit quantized (noisy) measurements, and, to the censored Tobit model, respectively. Depending on the situation, $g$ might be known or unspecified. In the statistics and econometrics literature, the measurement model in (1) is popular under the name *single-index model* and several aspects of it have been well-studied, e.g. [4,5,14,15][1].

**Structured Signals**. It is typical that the unknown signal $\mathbf{x}_0$ obeys some sort of *structure*. For instance, it might be sparse, i.e. only a few $k \ll n$, of its entries are non-zero; or, it might be that $\mathbf{x}_0 = \text{vec}(\mathbf{X}_0)$, where $\mathbf{X}_0 \in \mathbb{R}^{\sqrt{n} \times \sqrt{n}}$ is a matrix of low-rank $r \ll n$. To exploit this information it is typical to associate with the structure of $\mathbf{x}_0$ a properly chosen function $f : \mathbb{R}^n \to \mathbb{R}$, which we refer to as the *regularizer*. Of particular interest are *convex* and non-smooth such regularizers, e.g. the $\ell_1$-norm for sparse signals, the nuclear-norm for low-rank ones, etc. Please refer to [1,6,13] for further discussions.

**An Algorithm for Linear Measurements: The Generalized LASSO**. When the link function is *linear*, i.e. $g_i(x) = x + z_i$, perhaps the most popular way of estimating $\mathbf{x}_0$ is via solving the Generalized LASSO algorithm:

$$\hat{\mathbf{x}} := \arg\min_{\mathbf{x}} \|\mathbf{y} - \mathbf{A}\mathbf{x}\|_2 + \lambda f(\mathbf{x}). \tag{2}$$

Here, $\mathbf{A} = [\mathbf{a}_1, \mathbf{a}_2, \ldots, \mathbf{a}_m]^T \in \mathbb{R}^{m \times n}$ is the known measurement matrix and $\lambda > 0$ is a regularizer parameter. This is often referred to as the $\ell_2$-LASSO or the square-root-LASSO [3] to distinguish from the one solving $\min_{\mathbf{x}} \frac{1}{2} \|\mathbf{y} - \mathbf{A}\mathbf{x}\|_2^2 + \lambda f(\mathbf{x})$, instead. Our results can be accustomed to this latter version, but for concreteness, we restrict attention to (2) throughout. The acronym LASSO for (2) was introduced in [22] for the special case of $\ell_1$-regularization; (2) is a natural generalization to other kinds of structures and includes the group-LASSO [25], the fused-LASSO [23] as special cases. We often drop the term "Generalized" and refer to (2) simply as the LASSO.

One popular, measure of estimation performance of (2) is the squared-error $\|\hat{\mathbf{x}} - \mathbf{x}_0\|_2^2$. Recently, there have been significant advances on establishing tight bounds and even *precise* characterizations of this quantity, in the presence of linear measurements [2, 10, 16, 18, 19, 21]. Such precise results have been core to building a better understanding of the behavior of the LASSO, and, in particular, on the exact role played by the choice of the regularizer $f$ (in accordance with the structure of $\mathbf{x}_0$), by the number of measurements $m$, by the value of $\lambda$, etc.. In certain cases, they even provide us with useful insights into practical matters such as the tuning of the regularizer parameter.

**Using the LASSO for Non-linear Measurements?**. The LASSO is by nature tailored to a linear model for the measurements. Indeed, the first term of the objective function in (2) tries to fit $\mathbf{A}\mathbf{x}$ to the observed vector $\mathbf{y}$ presuming that this is of the form $y_i = \mathbf{a}_i^T \mathbf{x}_0 + \text{noise}$. Of course, no one stops us from continuing to use it even in cases where $y_i = g(\mathbf{a}_i^T \mathbf{x}_0)$ with $g$ being *non*-linear[2]. But, the question then becomes: Can there be any guarantees that the solution $\hat{\mathbf{x}}$ of the Generalized LASSO is still a good estimate of $\mathbf{x}_0$?

The question just posed was first studied back in the early 80's by Brillinger [5] who provided answers in the case of solving (2) without a regularizer term. This, of course, corresponds to standard Least Squares (LS). Interestingly, he showed that when the measurement vectors are Gaussian, then the LS solution is a consistent estimate of $\mathbf{x}_0$, up to a constant of proportionality $\mu$, which only depends on the link-function $g$. The result is sharp, but only under the assumption that the number of measurements $m$ grows large, while the signal dimension $n$ stays fixed, which was the typical setting of interest at the time. In the world of structured signals and high-dimensional measurements, the problem was only very recently revisited by Plan and Vershynin [17]. They consider a *constrained* version of the Generalized LASSO, in which the regularizer is essentially replaced by a constraint, and derive upper bounds on its performance. The bounds are not tight (they involve absolute constants), but they demonstrate some key features: i) the solution to the constrained LASSO $\hat{\mathbf{x}}$ is a good estimate of $\mathbf{x}_0$ up to the same constant of proportionality $\mu$ that appears in Brillinger's result. ii) Thus, $\|\hat{\mathbf{x}} - \mu\mathbf{x}_0\|_2^2$ is a natural measure of performance. iii) Estimation is possible even with $m < n$ measurements by taking advantage of the structure of $\mathbf{x}_0$.

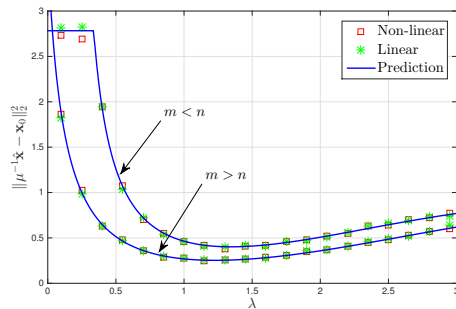

Figure 1: Squared error of the $\ell_1$-regularized LASSO with non-linear measurements ($\square$) and with corresponding linear ones ($\star$) as a function of the regularizer parameter $\lambda$; both compared to the asymptotic prediction. Here, $g_i(x) = \text{sign}(x + 0.3z_i)$ with $z_i \sim \mathcal{N}(0,1)$. The unknown signal $\mathbf{x}_0$ is of dimension $n = 768$ and has $\lceil 0.15n \rceil$ non-zero entries (see Sec. 2.2.2 for details). The different curves correspond to $\lceil 0.75n \rceil$ and $\lceil 1.2n \rceil$ number of measurements, respectively. Simulation points are averages over 20 problem realizations.

## 1.1 Summary of Contributions

Inspired by the work of Plan and Vershynin [17], and, motivated by recent advances on the precise analysis of the Generalized LASSO with linear measurements, this paper extends these latter results to the case of non-linear mesaurements. When the measurement matrix $\mathbf{A}$ has entries i.i.d. Gaussian (henceforth, we assume this to be the case without further reference), and the estimation performance is measured in a mean-squared-error sense, we are able to *precisely* predict the asymptotic behavior of the error. The derived expression accurately captures the role of the link function $g$, the particular structure of $\mathbf{x}_0$, the role of the regularizer $f$, and, the value of the regularizer parameter $\lambda$. Further, it holds for all values of $\lambda$, and for a wide class of functions $f$ and $g$.

Interestingly, our result shows in a very precise manner that in large dimensions, modulo the information about the magnitude of $\mathbf{x}_0$, the LASSO treats non-linear measurements exactly as if they were scaled and noisy linear measurements with scaling factor $\mu$ and noise variance $\sigma^2$ defined as

$$\mu := \mathbb{E}[\gamma g(\gamma)], \quad \text{and} \quad \sigma^2 := \mathbb{E}[(g(\gamma) - \mu\gamma)^2], \qquad \text{for } \gamma \sim \mathcal{N}(0,1), \tag{3}$$

where the expecation is with respect to both $\gamma$ and $g$. In particular, when $g$ is such that $\mu \neq 0$[3], then,

*the estimation performance of the Generalized LASSO with measurements of the form*
*$y_i = g_i(\mathbf{a}_i^T\mathbf{x}_0)$ is asymptotically the same as if the measurements were rather of the form*
*$y_i = \mu\mathbf{a}_i^T\mathbf{x}_0 + \sigma z_i$, with $\mu, \sigma^2$ as in (3) and $z_i$ standard gaussian noise.*

Recent analysis of the squared-error of the LASSO, when used to recover structured signals from noisy *linear* observations, provides us with either precise predictions (e.g. [2, 20]), or in other cases, with tight upper bounds (e.g. [10, 16]). Owing to the established relation between non-linear and (corresponding) linear measurements, such results also characterize the performance of the LASSO in the presence of nonlinearities. We remark that some of the error formulae derived here in the general context of non-linear measurements, have not been previously known even under the prism of linear measurements. Figure 1 serves as an illustration; the error with non-linear measurements matches well with the error of the corresponding linear ones and both are accurately predicted by our analytic expression.

Under the generic model in (1), which allows for $g$ to even be unspecified, $\mathbf{x}_0$ can, in principle, be estimated only up to a constant of proportionality [5, 15, 17]. For example, if $g$ is uknown then any information about the norm $\|\mathbf{x}_0\|_2$ could be absorbed in the definition of $g$. The same is true when $g(x) = \text{sign}(x)$, eventhough $g$ might be known here. In these cases, what becomes important is the *direction* of $\mathbf{x}_0$. Motivated by this, and, in order to simplify the presentation, we have assumed throughout that $\mathbf{x}_0$ *has unit Euclidean norm*[4], i.e. $\|\mathbf{x}_0\|_2 = 1$.

## 1.2 Discussion of Relevant Literature

**Extending an Old Result**. Brillinger [5] identified the asymptotic behavior of the estimation error of the LS solution $\hat{\mathbf{x}}_{LS} = (\mathbf{A}^T\mathbf{A})^{-1}\mathbf{A}^T\mathbf{y}$ by showing that, when $n$ (the dimension of $\mathbf{x}_0$) is fixed,

$$\lim_{m\to\infty} \sqrt{m}\|\hat{\mathbf{x}}_{LS} - \mu\mathbf{x}_0\|_2 = \sigma\sqrt{n}, \tag{4}$$

where $\mu$ and $\sigma^2$ are same as in (3). Our result can be viewed as a generalization of the above in several directions. First, we extend (4) to the regime where $m/n = \delta \in (1, \infty)$ and both grow large by showing that

$$\lim_{n\to\infty} \|\hat{\mathbf{x}}_{LS} - \mu\mathbf{x}_0\|_2 = \frac{\sigma}{\sqrt{\delta - 1}}. \tag{5}$$

Second, and most importantly, we consider solving the Generalized LASSO instead, to which LS is only a very special case. This allows versions of (5) where the error is finite even when $\delta < 1$ (e.g., see (8)). Note the additional challenges faced when considering the LASSO: i) $\hat{\mathbf{x}}$ no longer has a closed-form expression, ii) the result needs to additionally capture the role of $\mathbf{x}_0$, $f$, and, $\lambda$.

**Motivated by Recent Work**. Plan and Vershynin consider a *constrained* Generalized LASSO:

$$\hat{\mathbf{x}}_{\text{C-LASSO}} = \arg\min_{\mathbf{x}\in\mathcal{K}} \|\mathbf{y} - \mathbf{A}\mathbf{x}\|_2, \tag{6}$$

with $\mathbf{y}$ as in (1) and $\mathcal{K} \subset \mathbb{R}^n$ some known set (not necessarily convex). In its simplest form, their result shows that when $m \gtrsim D_{\mathcal{K}}(\mu\mathbf{x}_0)$ then with high probability,

$$\|\hat{\mathbf{x}}_{\text{C-LASSO}} - \mu\mathbf{x}_0\|_2 \lesssim \frac{\sigma\sqrt{D_{\mathcal{K}}(\mu\mathbf{x}_0)} + \zeta}{\sqrt{m}}. \tag{7}$$

Here, $D_{\mathcal{K}}(\mu\mathbf{x}_0)$ is the Gaussian width, a specific measure of complexity of the constrained set $\mathcal{K}$ when viewed from $\mu\mathbf{x}_0$. For our purposes, it suffices to remark that if $\mathcal{K}$ is properly chosen, and, if $\mu\mathbf{x}_0$ is on the boundary of $\mathcal{K}$, then $D_{\mathcal{K}}(\mu\mathbf{x}_0)$ is less than $n$. Thus, estimation is in principle is possible with $m < n$ measurements. The parameters $\mu$ and $\sigma$ that appear in (7) are the same as in (3) and $\zeta := \mathbb{E}[(g(\gamma) - \mu\gamma)^2\gamma^2]$. Observe that, in contrast to (4) and to the setting of this paper, the result in (7) is non-asymptotic. Also, it suggests the critical role played by $\mu$ and $\sigma$. On the other hand, (7) is only an upper bound on the error, and also, it suffers from unknown absolute proportionality constants (hidden in $\lesssim$).

Moving the analysis into an asymptotic setting, our work expands upon the result of [17]. First, we consider the regularized LASSO instead, which is more commonly used in practice. Most importantly, we improve the loose upper bounds into *precise* expressions. In turn, this proves in an exact manner the role played by $\mu$ and $\sigma^2$ to which (7) is only indicative. For a direct comparison with (7) we mention the following result which follows from our analysis (we omit the proof for brevity). Assume $\mathcal{K}$ is convex, $m/n = \delta \in (0, \infty)$, $D_{\mathcal{K}}(\mu\mathbf{x}_0)/n = \rho \in (0, 1]$ and $n \to \infty$. Also, $\delta > \rho$. Then, (7) yields an upper bound $C\sigma\sqrt{\rho/\delta}$ to the error, for some constant $C > 0$. Instead, we show

$$\|\hat{\mathbf{x}}_{\text{C-LASSO}} - \mu\mathbf{x}_0\|_2 \leq \sigma\frac{\sqrt{\rho}}{\sqrt{\delta - \rho}}. \tag{8}$$

**Precise Analysis of the LASSO With Linear Measurements**. The first *precise* error formulae were established in [2, 10] for the $\ell_2^2$-LASSO with $\ell_1$-regularization. The analysis was based on the the Approximate Message Passing (AMP) framework [9]. A more general line of work studies the problem using a recently developed framework termed the *Convex Gaussian Min-max Theorem* (CGMT) [19], which is a *tight* version of a classical Gaussian comparison inequality by Gordon [12]. The CGMT framework was initially used by Stojnic [18] to derive tight upper bounds on the constrained LASSO with $\ell_1$-regularization; [16] generalized those to general convex regularizers and also to the $\ell_2$-LASSO; the $\ell_2^2$-LASSO was studied in [21]. Those bounds hold for all values of SNR, but they become tight only in the high-SNR regime. A precise error expression for all values of SNR was derived in [20] for the $\ell_2$-LASSO with $\ell_1$-regularization under a gaussianity assumption on the distribution of the non-zero entries of $\mathbf{x}_0$. When measurements are linear, our Theorem 2.3 generalizes this assumption. Moreover, our Theorem 2.2 provides error predictions for regularizers going beyond the $\ell_1$-norm, e.g. $\ell_{1,2}$-norm, nuclear norm, which appear to be novel. When it comes to non-linear measurements, to the best of our knowledge, this paper is the first to derive asymptotically precise results on the performance of any LASSO-type program.

## 2 Results

### 2.1 Modeling Assumptions

**Unknown structured signal**. We let $\mathbf{x}_0 \in \mathbb{R}^n$ represent the unknown signal vector. We assume that $\mathbf{x}_0 = \bar{\mathbf{x}}_0/\|\bar{\mathbf{x}}_0\|_2$, with $\bar{\mathbf{x}}_0$ sampled from a probability density $p_{\bar{\mathbf{x}}_0}$ in $\mathbb{R}^n$. Thus, $\mathbf{x}_0$ is deterministically

of unit Euclidean-norm (this is mostly to simplify the presentation, see Footnote 4). Information about the structure of $\overline{\mathbf{x}}_0$ (and correspondingly of $\mathbf{x}_0$) is encoded in $p_{\overline{\mathbf{x}}_0}$. E.g., to study an $\overline{\mathbf{x}}_0$ which is sparse, it is typical to assume that its entries are i.i.d. $\overline{\mathbf{x}}_{0,i} \sim (1-\rho)\delta_0 + \rho q_{\overline{X}_0}$, where $\rho \in (0,1)$ becomes the normalized sparsity level, $q_{\overline{X}_0}$ is a scalar p.d.f. and $\delta_0$ is the Dirac delta function[5].

**Regularizer**. We consider *convex* regularizers $f : \mathbb{R}^n \to \mathbb{R}$.

**Measurement matrix**. The entries of $\mathbf{A} \in \mathbb{R}^{m \times n}$ are i.i.d. $\mathcal{N}(0,1)$.

**Measurements and Link-function**. We observe $\mathbf{y} = \vec{g}(\mathbf{A}\mathbf{x}_0)$ where $\vec{g}$ is a (possibly random) map from $\mathbb{R}^m$ to $\mathbb{R}^m$ and $\vec{g}(\mathbf{u}) = [g_1(u_1), \ldots, g_m(u_m)]^T$. Each $g_i$ is i.i.d. from a real valued random function $g$ for which $\mu$ and $\sigma^2$ are defined in (3). We assume that $\mu$ and $\sigma^2$ are nonzero and bounded.

**Asymptotics.** We study a linear asymptotic regime. In particular, we consider a sequence of problem instances $\{\overline{\mathbf{x}}_0^{(n)}, \mathbf{A}^{(n)}, f^{(n)}, m^{(n)}\}_{n \in \mathbb{N}}$ indexed by $n$ such that $\mathbf{A}^{(n)} \in \mathbb{R}^{m \times n}$ has entries i.i.d. $\mathcal{N}(0,1)$, $f^{(n)} : \mathbb{R}^n \to \mathbb{R}$ is proper convex, and, $m := m^{(n)}$ with $m = \delta n, \delta \in (0, \infty)$. We further require that the following conditions hold:

(a) $\overline{\mathbf{x}}_0^{(n)}$ is sampled from a probability density $p_{\overline{\mathbf{x}}_0}^{(n)}$ in $\mathbb{R}^n$ with one-dimensional marginals that are independent of $n$ and have bounded second moments. Furthermore, $n^{-1}\|\overline{\mathbf{x}}_0^{(n)}\|_2^2 \xrightarrow{P} \sigma_x^2 = 1$.

(b) For any $n \in \mathbb{N}$ and any $\|\mathbf{x}\|_2 \le C$, it holds $n^{-1/2} f(\mathbf{x}) \le c_1$ and $n^{-1/2} \max_{\mathbf{s} \in \partial f^{(n)}(\mathbf{x})} \|\mathbf{s}\|_2 \le c_2$, for constants $c_1, c_2, C \ge 0$ independent of $n$.

In (a), we used "$\xrightarrow{P}$" to denote convergence in probability as $n \to \infty$. The assumption $\sigma_x^2 = 1$ holds without loss of generality, and, is only necessary to simplify the presentation. In (b), $\partial f(\mathbf{x})$ denotes the subdifferential of $f$ at $\mathbf{x}$. The condition itself is no more than a normalization condition on $f$. Every such sequence $\{\overline{\mathbf{x}}_0^{(n)}, \mathbf{A}^{(n)}, f^{(n)}\}_{n \in \mathbb{N}}$ generates a sequence $\{\mathbf{x}_0^{(n)}, \mathbf{y}^{(n)}\}_{n \in \mathbb{N}}$ where $\mathbf{x}_0^{(n)} := \overline{\mathbf{x}}_0^{(n)} / \|\overline{\mathbf{x}}_0^{(n)}\|_2$ and $\mathbf{y}^{(n)} := \vec{g}^{(n)}(\mathbf{A}\mathbf{x}_0)$. When clear from the context, we drop the superscript $(n)$.

## 2.2 Precise Error Prediction

Let $\{\mathbf{x}_0^{(n)}, \mathbf{A}^{(n)}, f^{(n)}, \mathbf{y}^{(n)}\}_{n \in \mathbb{N}}$ be a sequence of problem instances that satisfying all the conditions above. With these, define the sequence $\{\hat{\mathbf{x}}^{(n)}\}_{n \in \mathbb{N}}$ of solutions to the corresponding LASSO problems for fixed $\lambda > 0$:

$$\hat{\mathbf{x}}^{(n)} := \min_{\mathbf{x}} \frac{1}{\sqrt{n}} \left\{ \|\mathbf{y}^{(n)} - \mathbf{A}^{(n)}\mathbf{x}\|_2 + \lambda f^{(n)}(\mathbf{x}) \right\}. \tag{9}$$

The main contribution of this paper is a precise evaluation of $\lim_{n \to \infty} \|\mu^{-1}\hat{\mathbf{x}}^{(n)} - \mathbf{x}_0^{(n)}\|_2^2$ with high probability over the randomness of $\mathbf{A}$, of $\mathbf{x}_0$, and of $g$.

### 2.2.1 General Result

To state the result in a general framework, we require a further assumption on $p_{\overline{\mathbf{x}}_0}^{(n)}$ and $f^{(n)}$. Later in this section we illustrate how this assumption can be naturally met. We write $f^*$ for the Fenchel's conjugate of $f$, i.e., $f^*(\mathbf{v}) := \sup_{\mathbf{x}} \mathbf{x}^T \mathbf{v} - f(\mathbf{x})$; also, we denote the Moreau envelope of $f$ at $\mathbf{v}$ with index $\tau$ to be $e_{f,\tau}(\mathbf{v}) := \min_{\mathbf{x}} \{\frac{1}{2}\|\mathbf{v} - \mathbf{x}\|_2^2 + \tau f(\mathbf{x})\}$.

**Assumption 1.** *We say Assumption 1 holds if for all non-negative constants $c_1, c_2, c_3 \in \mathbb{R}$ the point-wise limit of $\frac{1}{n} e_{\sqrt{n}(f^*)^{(n)}, c_3} (c_1 \mathbf{h} + c_2 \overline{\mathbf{x}}_0)$ exists with probability one over $\mathbf{h} \sim \mathcal{N}(0, \mathbf{I}_n)$ and $\overline{\mathbf{x}}_0 \sim p_{\overline{\mathbf{x}}_0}^{(n)}$. Then, we denote the limiting value as $F(c_1, c_2, c_3)$.*

**Theorem 2.1** (Non-linear=Linear)**.** *Consider the asymptotic setup of Section 2.1 and let Assumption 1 hold. Recall $\mu$ and $\sigma^2$ as in (3) and let $\hat{\mathbf{x}}$ be the minimizer of the Generalized LASSO in (9) for fixed $\lambda > 0$ and for measurements given by (1). Further let $\hat{\mathbf{x}}^{lin}$ be the solution to the Generalized LASSO when used with* linear *measurements of the form $\mathbf{y}^{lin} = \mathbf{A}(\mu\mathbf{x}_0) + \sigma\mathbf{z}$, where $\mathbf{z}$ has entries i.i.d. standard normal. Then, in the limit of $n \to \infty$, with probability one,*

$$\|\hat{\mathbf{x}} - \mu\mathbf{x}_0\|_2^2 = \|\hat{\mathbf{x}}^{lin} - \mu\mathbf{x}_0\|_2^2.$$

Theorem 2.1 relates in a very precise manner the error of the Generalized LASSO under non-linear measurements to the error of the same algorithm when used under appropriately scaled noisy linear measurements. Theorem 2.2 below, derives an asymptotically exact expression for the error.

**Theorem 2.2** (Precise Error Formula). *Under the same assumptions of Theorem 2.1 and $\delta := m/n$, it holds, with probability one,*

$$\lim_{n \to \infty} \|\hat{\mathbf{x}} - \mu \mathbf{x}_0\|_2^2 = \alpha_*^2,$$

*where $\alpha_*$ is the unique optimal solution to the convex program*

$$\max_{\substack{0 \le \beta \le 1 \\ \tau \ge 0}} \min_{\alpha \ge 0} \beta \sqrt{\delta} \sqrt{\alpha^2 + \sigma^2} - \frac{\alpha \tau}{2} + \frac{\mu^2 \tau}{2\alpha} - \frac{\alpha \lambda^2}{\tau} F\left( \frac{\beta}{\lambda}, \frac{\mu \tau}{\lambda \alpha}, \frac{\tau}{\lambda \alpha} \right). \tag{10}$$

*Also, the optimal cost of the LASSO in (9) converges to the optimal cost of the program in (10).*

Under the stated conditions, Theorem 2.2 proves that the limit of $\|\hat{\mathbf{x}} - \mu \mathbf{x}_0\|_2$ exists and is equal to the *unique* solution of the optimization program in (10). Notice that this is a *deterministic* and *convex* optimization, which only involves three *scalar* optimization variables. Thus, the optimal $\alpha_*$ can, in principle, be efficiently numerically computed. In many specific cases of interest, with some extra effort, it is possible to yield simpler expressions for $\alpha_*$, e.g. see Theorem 2.3 below. The role of the normalized number of measurement $\delta = m/n$, of the regularizer parameter $\lambda$, and, that of $g$, through $\mu$ and $\sigma^2$, are explicit in (10); the structure of $\mathbf{x}_0$ and the choice of the regularizer $f$ are implicit in $F$. Figures 1-2 illustrate the accuracy of the prediction of the theorem in a number of different settings. The proofs of both the Theorems are deferred to Appendix A. In the next sections, we specialize Theorem 2.2 to the cases of sparse, group-sparse and low-rank signal recovery.

### 2.2.2 Sparse Recovery

Assume each entry $\overline{\mathbf{x}}_{0,i}, i = 1, \ldots, n$ is sampled i.i.d. from a distribution

$$p_{\overline{X}_0}(x) = (1 - \rho) \cdot \delta_0(x) + \rho \cdot q_{\overline{X}_0}(x), \tag{11}$$

where $\delta_0$ is the delta Dirac function, $\rho \in (0, 1)$ and $q_{\overline{X}_0}$ a probability density function with second moment normalized to $1/\rho$ so that condition (a) of Section 2.1 is satisfied. Then, $\mathbf{x}_0 = \overline{\mathbf{x}}_0/\|\overline{\mathbf{x}}_0\|_2$ is $\rho n$-sparse on average and has unit Euclidean norm. Letting $f(\mathbf{x}) = \|\mathbf{x}\|_1$ also satisfies condition (b). Let us now check Assumption 1. The Fenchel's conjugate of the $\ell_1$-norm is simply the indicator function of the $\ell_\infty$ unit ball. Hence, without much effort,

$$\frac{1}{n} \mathrm{e}_{\sqrt{n}(f^*)^{(n)}, c_3}(c_1 \mathbf{h} + c_2 \overline{\mathbf{x}}_0) = \frac{1}{2n} \sum_{i=1}^{n} \min_{|v_i| \le 1} (v_i - (c_1 \mathbf{h}_i + c_2 \overline{\mathbf{x}}_{0,i}))^2$$

$$= \frac{1}{2n} \sum_{i=1}^{n} \eta^2(c_1 \mathbf{h}_i + c_2 \overline{\mathbf{x}}_{0,i}; 1), \tag{12}$$

where we have denoted

$$\eta(x; \tau) := (x/|x|) (|x| - \tau)_+ \tag{13}$$

for the soft thresholding operator. An application of the weak law of large numbers to see that the limit of the expression in (12) equals $F(c_1, c_2, c_3) := \frac{1}{2} \mathbb{E}\left[\eta^2(c_1 h + c_2 \overline{X}_0; 1)\right]$, where the expectation is over $h \sim \mathcal{N}(0, 1)$ and $\overline{X}_0 \sim p_{\overline{X}_0}$. With all these, Theorem 2.2 is applicable. We have put extra effort in order to obtain the following equivalent but more insightful characterization of the error, as stated below and proved in Appendix B.

**Theorem 2.3** (Sparse Recovery). *If $\delta > 1$, then define $\lambda_{crit} = 0$. Otherwise, let $\lambda_{crit}, \kappa_{crit}$ be the unique pair of solutions to the following set of equations:*

$$\begin{cases} \kappa^2 \delta = \sigma^2 + \mathbb{E}\left[(\eta(\kappa h + \mu \overline{X}_0; \kappa \lambda) - \mu \overline{X}_0)^2\right], & \tag{14} \\ \kappa \delta = \mathbb{E}[(\eta(\kappa h + \mu \overline{X}_0; \kappa \lambda) \cdot h)], & \tag{15} \end{cases}$$

*where $h \sim \mathcal{N}(0, 1)$ and is independent of $\overline{X}_0 \sim p_{\overline{X}_0}$. Then, for any $\lambda > 0$, with probability one,*

$$\lim_{n \to \infty} \|\hat{\mathbf{x}} - \mu \mathbf{x}_0\|_2^2 = \begin{cases} \delta \kappa_{crit}^2 - \sigma^2 & , \lambda \le \lambda_{crit}, \\ \delta \kappa_*^2(\lambda) - \sigma^2 & , \lambda \ge \lambda_{crit}, \end{cases}$$

*where $\kappa_*^2(\lambda)$ is the unique solution to (14).*

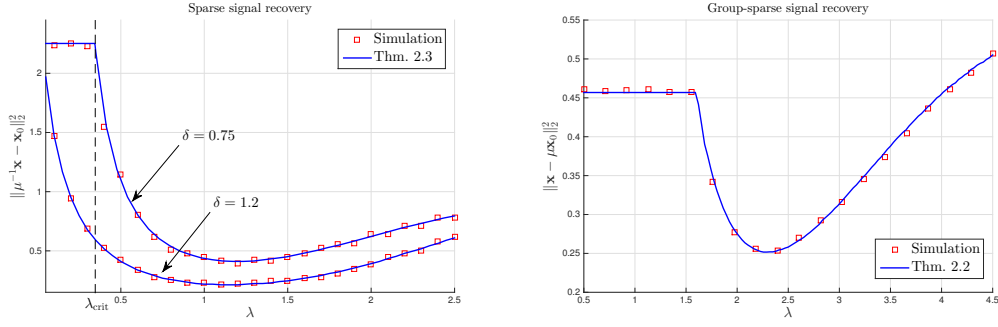

Figure 2: Squared error of the LASSO as a function of the regularizer parameter compared to the asymptotic predictions. Simulation points represent averages over 20 realizations. (a) Illustration of Thm. 2.3 for $g(x) = \text{sign}(x)$, $n = 512$, $p_{\overline{X}_0}(+1) = p_{\overline{X}_0}(+1) = 0.05$, $p_{\overline{X}_0}(+1) = 0.9$ and two values of $\delta$, namely 0.75 and 1.2. (b) Illustration of Thm. 2.2 for $\mathbf{x}_0$ being group-sparse as in Section 2.2.3 and $g_i(x) = \text{sign}(x + 0.3z_i)$. In particular, $\mathbf{x}_0$ is composed of $t = 512$ blocks of block size $b = 3$. Each block is zero with probability 0.95, otherwise its entries are i.i.d. $\mathcal{N}(0,1)$. Finally, $\delta = 0.75$.

Figures 1 and 2(a) validate the prediction of the theorem, for different signal distributions, namely $q_{\overline{X}_0}$ being Gaussian and Bernoulli, respectively. For the case of compressed ($\delta < 1$) measurements, observe the two different regimes of operation, one for $\lambda \leq \lambda_{\text{crit}}$ and the other for $\lambda \geq \lambda_{\text{crit}}$, precisely as they are predicted by the theorem (see also [16, Sec. 8]). The special case of Theorem 2.3 for which $q_{\overline{X}_0}$ is Gaussian has been previously studied in [20]. Otherwise, to the best of our knowledge, this is the first precise analysis result for the $\ell_2$-LASSO stated in that generality. Analogous result, but via different analysis tools, has only been known for the $\ell_2^2$-LASSO as appears in [2].

### 2.2.3 Group-Sparse Recovery

Let $\overline{\mathbf{x}}_0 \in \mathbb{R}^n$ be composed of $t$ non-overlapping blocks of constant size $b$ each such that $n = t \cdot b$. Each block $[\overline{\mathbf{x}}_0]_i, i = 1, \ldots, t$ is sampled i.i.d. from a probability density in $\mathbb{R}^b$: $p_{\overline{X}_0}(\mathbf{x}) = (1 - \rho) \cdot \delta_0(\mathbf{x}) + \rho \cdot q_{\overline{X}_0}(\mathbf{x}), \mathbf{x} \in \mathbb{R}^b$, where $\rho \in (0,1)$. Thus, $\overline{\mathbf{x}}_0$ is $\rho t$-block-sparse on average. We operate in the regime of linear measurements $m/n = \delta \in (0,\infty)$. As is common we use the $\ell_{1,2}$-norm to induce block-sparsity, i.e., $f(\mathbf{x}) = \sum_{i=1}^t \|[\overline{\mathbf{x}}_0]_i\|_2$; with this, (9) is often referred to as group-LASSO in the literature [25]. It is not hard to show that Assumption 1 holds with $F(c_1, c_2, c_3) := \frac{1}{2b}\mathbb{E}\left[\|\vec{\eta}(c_1 \mathbf{h} + c_2\overline{X}_0; 1)\|_2^2\right]$, where $\vec{\eta}(\mathbf{x}; \tau) = \mathbf{x}/\|\mathbf{x}\| \left(\|\mathbf{x}\|_2 - \tau\right)_+, \mathbf{x} \in \mathbb{R}^b$ is the vector soft thresholding operator and $h \sim \mathcal{N}(0, \mathbf{I}_b), \overline{X}_0 \sim p_{\overline{X}_0}$ and are independent. Thus Theorem 2.2 is applicable in this setting; Figure 2(b) illustrates the accuracy of the prediction.

### 2.2.4 Low-rank Matrix Recovery

Let $\overline{\mathbf{X}}_0 \in \mathbb{R}^{d \times d}$ be an unknown matrix of rank $r$, in which case, $\overline{\mathbf{x}}_0 = \text{vec}(\overline{\mathbf{X}}_0)$ with $n = d^2$. Assume $m/d^2 = \delta \in (0, \infty)$ and $r/d = \rho \in (0, 1)$. As usual in this setting, we consider nuclear-norm regularization; in particular, we choose $f(\mathbf{x}) = \sqrt{d}\|\mathbf{X}\|_*$. Each subgradient $\mathbf{S} \in \partial f(\mathbf{X})$ then satisfies $\|\mathbf{S}\|_F \leq d$ in agreement with assumption (b) of Section 2.1. Furthermore, for this choice of regularizer, we have

$$\frac{1}{n}\text{e}_{\sqrt{n}(f^*)^{(n)}, c_3}\left(c_1\mathbf{H} + c_2\overline{\mathbf{X}}_0\right) = \frac{1}{2d^2}\min_{\|\mathbf{V}\|_2 \leq \sqrt{d}}\|\mathbf{V} - (c_1\mathbf{H} + c_2\overline{\mathbf{X}}_0)\|_F^2$$

$$= \frac{1}{2d}\min_{\|\mathbf{V}\|_2 \leq 1}\|\mathbf{V} - d^{-1/2}(c_1\mathbf{H} + c_2\overline{\mathbf{X}}_0)\|_F^2 = \frac{1}{2d}\sum_{i=1}^d \eta^2\left(\text{s}_i\left(d^{-1/2}(c_1\mathbf{H} + c_2\overline{\mathbf{X}}_0)\right); 1\right),$$

where $\eta(\cdot; \cdot)$ is as in (13), $\text{s}_i(\cdot)$ denotes the $i^{\text{th}}$ singular value of its argument and $\mathbf{H} \in \mathbb{R}^{d \times d}$ has entries $\mathcal{N}(0, 1)$. If conditions are met such that the empirical distribution of the singular values of (the sequence of random matrices) $c_1\mathbf{H} + c_2\overline{\mathbf{X}}_0$ converges asymptotically to a limiting distribution, say $q(c_1, c_2)$, then $F(c_1, c_2, c_3) := \frac{1}{2}\mathbb{E}_{x \sim q(c_1, c_2)}\left[\eta^2(x; 1)\right]$, and Theorem 2.1–2.2 apply. For instance, this will be the case if $d^{-1/2}\overline{\mathbf{X}}_0 = \mathbf{USV}^t$, where $\mathbf{U}, \mathbf{V}$ unitary matrices and $\mathbf{S}$ is a diagonal matrix

whose entries have a given marginal distribution with bounded moments (in particular, independent of $d$). We leave the details and the problem of (numerically) evaluating $F$ for future work.

## 2.3 An Application to $q$-bit Compressive Sensing

### 2.3.1 Setup

Consider recovering a *sparse* unknown signal $\mathbf{x}_0 \in \mathbb{R}^n$ from scalar q-bit quantized linear measurements. Let $\mathbf{t} := \{t_0 = 0, t_1, \ldots, t_{L-1}, t_L = +\infty\}$ represent a (symmetric with respect to 0) set of decision thresholds and $\boldsymbol{\ell} := \{\pm\ell_1, \pm\ell_2, \ldots, \pm\ell_L\}$ the corresponding representation points, such that $L = 2^{q-1}$. Then, quantization of a real number $x$ into $q$-bits can be represented as

$$\mathcal{Q}_q(x, \boldsymbol{\ell}, \mathbf{t}) = \text{sign}(x) \sum_{i=1}^{L} \ell_i \mathbf{1}_{\{t_{i-1} \leq |x| \leq t_i\}},$$

where $\mathbf{1}_\mathcal{S}$ is the indicator function of a set $\mathcal{S}$. For example, 1-bit quantization with level $\ell$ corresponds to $Q_1(x, \ell) = \ell \cdot \text{sign}(x)$. The measurement vector $\mathbf{y} = [y_1, y_2 \ldots, y_m]^T$ takes the form

$$y_i = \mathcal{Q}_q(\mathbf{a}_i^T \mathbf{x}_0, \boldsymbol{\ell}, \mathbf{t}), \quad i = 1, 2, \ldots, m, \tag{16}$$

where $\mathbf{a}_i^T$'s are the rows of a measurement matrix $\mathbf{A} \in \mathbb{R}^{m \times n}$, which is henceforth assumed i.i.d. standard Gaussian. We use the LASSO to obtain an estimate $\hat{\mathbf{x}}$ of $\mathbf{x}_0$ as

$$\hat{\mathbf{x}} := \arg \min_{\mathbf{x}} \|\mathbf{y} - \mathbf{A}\mathbf{x}\|_2 + \lambda \|\mathbf{x}\|_1. \tag{17}$$

Henceforth, we assume for simplicity that $\|\mathbf{x}_0\|_2 = 1$. Also, in our case, $\mu$ is known since $g = Q_q$ is known; thus, is reasonable to scale the solution of (17) as $\mu^{-1}\hat{\mathbf{x}}$ and consider the error quantity $\|\mu^{-1}\hat{\mathbf{x}} - \mathbf{x}_0\|_2$ as a measure of estimation performance. Clearly, the error depends (besides others) on the number of bits $q$, on the choice of the decision thresholds $\mathbf{t}$ and on the quantization levels $\boldsymbol{\ell}$. An interesting question of practical importance becomes how to optimally choose these to achieve less error. As a running example for this section, we seek optimal quantization thresholds and corresponding levels

$$(\mathbf{t}_*, \boldsymbol{\ell}_*) = \arg \min_{\mathbf{t}, \boldsymbol{\ell}} \|\mu^{-1}\hat{\mathbf{x}} - \mathbf{x}_0\|_2, \tag{18}$$

while keeping all other parameters such as the number of bits $q$ and of measurements $m$ fixed.

### 2.3.2 Consequences of Precise Error Prediction

Theorem 2.1 shows that $\|\mu^{-1}\hat{\mathbf{x}} - \mathbf{x}_0\|_2 = \|\hat{\mathbf{x}}^{\text{lin}} - \mathbf{x}_0\|_2$, where $\hat{\mathbf{x}}^{\text{lin}}$ is the solution to (17), but only, this time with a measurement vector $\mathbf{y}^{\text{lin}} = \mathbf{A}\mathbf{x}_0 + \frac{\sigma}{\mu}\mathbf{z}$, where $\mu, \sigma$ as in (20) and $\mathbf{z}$ has entries i.i.d. standard normal. Thus, lower values of the ratio $\sigma^2/\mu^2$ correspond to lower values of the error and the design problem posed in (18) is equivalent to the following simplified one:

$$(\mathbf{t}_*, \boldsymbol{\ell}_*) = \arg \min_{\mathbf{t}, \boldsymbol{\ell}} \frac{\sigma^2(\mathbf{t}, \boldsymbol{\ell})}{\mu^2(\mathbf{t}, \boldsymbol{\ell})}. \tag{19}$$

To be explicit, $\mu$ and $\sigma^2$ above can be easily expressed from (3) after setting $g = \mathcal{Q}_q$ as follows:

$$\mu := \mu(\boldsymbol{\ell}, \mathbf{t}) = \sqrt{\frac{2}{\pi}} \sum_{i=1}^{L} \ell_i \cdot \left( e^{-t_{i-1}^2/2} - e^{-t_i^2/2} \right) \quad \text{and} \quad \sigma^2 := \sigma^2(\boldsymbol{\ell}, \mathbf{t}) := \tau^2 - \mu^2, \tag{20}$$

where, $\tau^2 := \tau^2(\boldsymbol{\ell}, \mathbf{t}) = 2 \sum_{i=1}^{L} \ell_i^2 \cdot (Q(t_{i-1}) - Q(t_i))$ and $Q(x) = \frac{1}{\sqrt{2\pi}} \int_x^\infty \exp(-u^2/2) \mathrm{d}u$.

### 2.3.3 An Algorithm for Finding Optimal Quantization Levels and Thresholds

In contrast to the initial problem in (18), the optimization involved in (19) is explicit in terms of the variables $\boldsymbol{\ell}$ and $\mathbf{t}$, but, is still hard to solve in general. Interestingly, we show in Appendix C that the popular Lloyd-Max (LM) algorithm can be an effective algorithm for solving (19), since the values to which it converges are stationary points of the objective in (19). Note that this is not a directly obvious result since the classical objective of the LM algorithm is minimizing the quantity $\mathbb{E}[\|\mathbf{y} - \mathbf{A}\mathbf{x}_0\|_2^2]$ rather than $\mathbb{E}[\|\mu^{-1}\hat{\mathbf{x}} - \mathbf{x}_0\|_2^2]$.

## Footnotes

*This work was supported in part by the National Science Foundation under grants CNS-0932428, CCF-1018927, CCF-1423663 and CCF-1409204, by a grant from Qualcomm Inc., by NASA's Jet Propulsion Laboratory through the President and Directors Fund, by King Abdulaziz University, and by King Abdullah University of Science and Technology.

[1] The single-index model is a classical topic and can also be regarded as a special case of what is known as *sufficient dimension reduction* problem. There is extensive literature on both subjects; unavoidably, we only refer to the directly relevant works here.

[2] Note that the Generalized LASSO in (2) does not assume knowledge of $g$. All that is assumed is the availability of the measurements $y_i$. Thus, the link-function might as well be unknown or unspecified.

[3]This excludes for example link functions $g$ that are even, but also some other not so obvious cases [11, Sec. 2.2]. For a few special cases, e.g. sparse recovery with binary measurements $y_i$ [24], different methodologies than the LASSO have been recently proposed that do not require $\mu = 0$.

[4]In [17, Remark 1.8], they note that their results can be easily generalized to the case when $\|\mathbf{x}_0\|_2 \neq 1$ by simply redifining $\bar{g}(x) = g(\|\mathbf{x}_0\|_2 x)$ and accordingly adjusting the values of the parameters $\mu$ and $\sigma^2$ in (3). The very same argument is also true in our case.

[5]Such models have been widely used in the relevant literature, e.g. [7,8,10]. In fact, the results here continue to hold as long as the marginal distribution of $\overline{\mathbf{x}}_0$ converges to a given distribution (as in [2]).

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
