[Supplementary Material · SupplementaryMaterial.pdf]

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

# A Proof of Theorems 2.1 & 2.2

## A.1 Theorem 2.2

We start with the proof of Theorem 2.2. Theorem 2.1 will follow as a direct corollary of this result.

Assume a sequence of problem instances as described in Section 2.1. To keep notation simple, we simply use $\|\mathbf{v}\|$ (rather than $\|\mathbf{v}\|_2$) for the Euclidean norm of $\mathbf{v}$ and we shall also drop the superscript $(n)$ when referring to elements of the sequence. Thus, we write

$$\hat{\mathbf{x}} = \arg\min_{\mathbf{x}} \frac{1}{\sqrt{n}}\|\vec{g}(\mathbf{A}\mathbf{x}_0) - \mathbf{A}\mathbf{x}\| + \frac{\lambda}{\sqrt{n}}f(\mathbf{x}), \tag{21}$$

but it is to be understood that the above actually produces a sequence of solutions $\hat{\mathbf{x}}^{(n)}$ indexed by $n$. Our goal is to characterize the nontrivial limiting behavior of $\|\hat{\mathbf{x}} - \mu\mathbf{x}_0\|$.

We start with a simple but useful change of variables $\mathbf{w} := \mathbf{x} - \mu\mathbf{x}_0$, to directly get a handle on the error vector $\mathbf{w}$. Then, (21) becomes:

$$\hat{\mathbf{w}} := \arg\min_{\mathbf{w}} \frac{1}{\sqrt{n}}\|\vec{g}(\mathbf{A}\mathbf{x}_0) - \mu\mathbf{A}\mathbf{x}_0 - \mathbf{A}\mathbf{w}\| + \frac{\lambda}{\sqrt{n}}f(\mu\mathbf{x}_0 + \mathbf{w})$$

$$= \arg\min_{\mathbf{w}} \max_{\|\mathbf{u}\|\leq 1} \frac{1}{\sqrt{n}}(-\mathbf{u}^T\mathbf{A}\mathbf{w} + \mathbf{u}^T(\vec{g}(\mathbf{A}\mathbf{x}_0) - \mu\mathbf{A}\mathbf{x}_0)) + \frac{\lambda}{\sqrt{n}}f(\mu\mathbf{x}_0 + \mathbf{w}), \tag{22}$$

where the second line follows after using the fact $\|\mathbf{v}\| = \max_{\|\mathbf{u}\|\leq 1} \mathbf{u}^T\mathbf{v}$.

### A.1.1 A Key Decomposition

The first key step in the proof is a trick adapted from the proofs of [20, Lem. 4.3] and [21, Thm. 1.3]. Until further notice, we condition on $\mathbf{x}_0$. Also, we repeatedly make use of the assumption that $\|\mathbf{x}_0\| = 1$ without direct reference. The trick amounts to decomposing each measurement vector $\mathbf{a}_i$ in its projection on the direction of $\mathbf{x}_0$ and its orthogonal complement. Denoting $\mathbf{P}^\perp = (\mathbf{I} - \mathbf{x}_0\mathbf{x}_0^T)$ for the projector onto the orthogonal complement of the span of $\mathbf{x}_0$ (recall $\|\mathbf{x}_0\|_2 = 1$), we have $\mathbf{a}_i^T = (\mathbf{a}_i^T\mathbf{x}_0)\mathbf{x}_0^T + \mathbf{a}_i^T\mathbf{P}^\perp$, or, in matrix form:

$$\mathbf{A} = (\mathbf{A}\mathbf{x}_0)\mathbf{x}_0^T + \mathbf{A}\mathbf{P}^\perp.$$

Then, (22) becomes:

$$\min_{\mathbf{w}} \max_{\|\mathbf{u}\|\leq 1} \frac{1}{\sqrt{n}}\left\{-\mathbf{u}^T\mathbf{A}\mathbf{P}^\perp\mathbf{w} + \mathbf{u}^T(\vec{g}(\mathbf{A}\mathbf{x}_0) - \mu\mathbf{A}\mathbf{x}_0 - (\mathbf{A}\mathbf{x}_0)\mathbf{x}_0^T\mathbf{w})\right\} + \frac{\lambda}{\sqrt{n}}f(\mu\mathbf{x}_0 + \mathbf{w}). \tag{23}$$

Using the Gaussianity assumption on the entries of $\mathbf{A}$ it is straightforward to show that $\mathbf{P}^\perp\mathbf{a}_i$ is *independent* of $\mathbf{a}_i^T\mathbf{x}_0$, for all $i = 1, \ldots, m$. Also, conditioned on $\mathbf{a}_i^T\mathbf{x}_0$, $\mathbf{P}^\perp\mathbf{a}_i$ is independent of $(\vec{g}_i(\mathbf{a}_i^T\mathbf{x}_0) - \mu\mathbf{a}_i^T\mathbf{x}_0)$ since the latter only depends on $\mathbf{a}_i$ through $\mathbf{a}_i^T\mathbf{x}_0$. Combining those, it follows that $\mathbf{P}^\perp\mathbf{a}_i$ is also *independent* of $(\vec{g}_i(\mathbf{a}_i^T\mathbf{x}_0) - \mu\mathbf{a}_i^T\mathbf{x}_0)$, [20, pg. 13]. Overall, $\mathbf{A}\mathbf{P}^\perp\mathbf{w}$ is independent of the rest terms in in (23). This shows that the objective function of (23) is distributed identically even after replacing the $\mathbf{A}\mathbf{P}^\perp\mathbf{w}$ with $\mathbf{G}\mathbf{P}^\perp\mathbf{w}$, where $\mathbf{G}$ is an independent copy of $\mathbf{A}$. After all these, (23) is identically distributed with the following:

$$\min_{\mathbf{w}} \max_{\|\mathbf{u}\|\leq 1} \frac{1}{\sqrt{n}}\{-\mathbf{u}^T\mathbf{G}\mathbf{P}^\perp\mathbf{w} + \mathbf{u}^T(\mathbf{z_e} - (\mathbf{x}_0^T\mathbf{w})\mathbf{e})\} + \frac{\lambda}{\sqrt{n}}f(\mu\mathbf{x}_0 + \mathbf{w}). \tag{24}$$

where $\mathbf{G}$ and $\mathbf{e} := \mathbf{A}\mathbf{x}_0$ have entries i.i.d. standard normal and are independent of each other. Also, $\mathbf{z_e} := \vec{g}(\mathbf{e}) - \mu\mathbf{e}$ for convenience.

### A.1.2 Applying the CGMT

After the decomposition step in the previous section, we have transformed the initial problem to that of analyzing the (probabilistically) equivalent one in (24). In particular, we wish to evaluate the limiting behavior of $\|\hat{\mathbf{w}}\|$, i.e. the norm of the minimizer of the optimization in (24). The

analysis is possible thanks to the Convex Gaussian Min-max Theorem (CGMT) [28, Thm. 3], which is a stronger version of the classical result of Gordon [13] in the presence of additional convexity assumptions. According to the CGMT, the analysis of a Primary Optimization (PO) problem that is of the form

$$\min_{\mathbf{v}\in\mathcal{S}_{\mathbf{v}}} \max_{\mathbf{u}\in\mathcal{S}_{\mathbf{u}}} \mathbf{u}^T\mathbf{G}\mathbf{v} + \psi(\mathbf{v},\mathbf{u}), \tag{25}$$

with $\mathbf{G}$ being i.i.d. Gaussian, $\mathcal{S}_{\mathbf{v}}, \mathcal{S}_{\mathbf{u}}$ convex, compact sets and $\psi$ a convex-concave function, can be carried out via analyzing a corresponding Auxiliary Optimization problem (AO) defined as

$$\min_{\mathbf{v}\in\mathcal{S}_{\mathbf{v}}} \max_{\mathbf{u}\in\mathcal{S}_{\mathbf{u}}} \|\mathbf{v}\|\mathbf{g}^T\mathbf{u} + \|\mathbf{u}\|\mathbf{h}^T\mathbf{v} + \psi(\mathbf{v},\mathbf{u}). \tag{26}$$

Here, and onwards, $\mathbf{g}$ and $\mathbf{h}$ are i.i.d. standard Gaussian vectors of appropriate size. In (24) identify the bilinear term $\mathbf{u}^T\mathbf{G}\mathbf{P}^{\perp}\mathbf{w}$ and note that the rest of the objective function is convex in $\mathbf{w}$ (recall that $f$ is convex), and, linear (thus, concave) in $\mathbf{u}$. Overall, this is in the appropriate format of a (PO) problem as in (25) modulo the extra factor $\mathbf{P}^{\perp}$ in the bilinear term. It is straightforward to show that this extra factor only requires a natural change of the corresponding terms in the (AO) problem as follows: $\|\mathbf{P}^{\perp}\mathbf{w}\|\mathbf{g}^T\mathbf{u} + \|\mathbf{u}\|_2\mathbf{h}^T\mathbf{P}^{\perp}\mathbf{w}$. With this minor modification, the CGMT continues to hold. A remaining technical caveat is that the minimization over $\mathbf{w}$ in it appears unconstrained. For this, we assume that the minimizer of (24) satisfies $\|\hat{\mathbf{w}}\| \leq K_{\mathbf{w}}$ for sufficiently large constant $K_{\mathbf{w}} > 0$ *independent of* $n$. If our assumption is valid, then by the end of the proof we will have identified a quantity $\alpha_* > 0$ to which $\|\hat{\mathbf{w}}\|$ converges; If $\alpha_*$ turns out to be independent of the choice of $K_{\mathbf{w}}$, then we may explicitly choose $K_{\mathbf{w}} = 2\alpha_*$ (say) and $\alpha_*$ is the true limit; on the other hand, if $\alpha_*$ turns out to depend on $K_{\mathbf{w}}$, this means that we could have chosen $K_{\mathbf{w}}$ arbitrarily large in the first place, and so the true limit diverges. Thus, assuming that $\|\hat{\mathbf{w}}\|$ the minimization in (24) is not affected by imposing the constraint $\|\mathbf{w}\| \leq K_{\mathbf{w}}$. With these, we can write the corresponding (AO) problem as

$$\tilde{\mathbf{w}} = \arg\min_{\|\mathbf{w}\|\leq K_{\mathbf{w}}} \max_{\|\mathbf{u}\|\leq 1} \frac{1}{\sqrt{n}}\{\|\mathbf{P}^{\perp}\mathbf{w}\|\mathbf{g}^T\mathbf{u} - \|\mathbf{u}\|\mathbf{h}^T\mathbf{P}^{\perp}\mathbf{w} + \mathbf{u}^T(\mathbf{z_e} - (\mathbf{x}_0^T\mathbf{w})\mathbf{e})\} + \frac{\lambda}{\sqrt{n}}f(\mu\mathbf{x}_0 + \mathbf{w}).$$
$$\tag{27}$$

We will see that analyzing this problem, with the goal of determining the converging value of the magnitude of its minimizer $\tilde{\mathbf{w}}$, is simpler than analyzing the (PO) (and certainly so of the one we started with in (22)). The CGMT essentially shows that $\|\tilde{\mathbf{w}}\|$ converges to the same value as $\|\hat{\mathbf{w}}\|$. Recall $\hat{\mathbf{w}}$ being the minimizer of the (PO) and the goal of Theorem 2.2 being to evaluate the converging value of its magnitude.

### A.1.3 Analysis of the Auxiliary Optimization

The goal of this section is that of analyzing the (AO) problem in (27). In particular, we will prove that (i) the optimal cost of the (AO) problem converges to the optimal cost of the deterministic optimization in (10), which involves three scalar optimization variables $\alpha, \beta, \tau$, (ii) the max-min problem in (10) is *strongly* convex in $\alpha$ and jointly concave in $\beta, \tau$, (iii) $\|\tilde{\mathbf{w}}\|$ converges to the unique optima $\alpha_*$ in (10). With these, the claim of the Theorem follows by [28, Thm. 1] (also, see [27, Cor. A.1]), as previously discussed.

The analysis requires several steps. The randomness in (27) is over $\mathbf{e}, \mathbf{g}, \mathbf{h}, \mathbf{x}_0$ and possibly the link function $g$; at each step we condition on all but a subset of these and identify convergence of the objective function of the (AO) with respect to the remaining. Pointwise convergence (with respect to the involved optimization variables) needs to be turned into uniform convergence to guarantee that not only the objective function, but also the min/max value and the optimizer converge appropriately. (Strong) convexity of the objective will turn out to be crucial for the latter.

**Introducing the Frenchel conjugate.** To begin with, let us rewrite the (AO) problem above by expressing $f$ in terms of its Frenchel conjugate, i.e.

$$f(\mathbf{x}) = \sup_{\bar{\mathbf{v}}} \bar{\mathbf{v}}^T\mathbf{x} - f^*(\bar{\mathbf{v}}) = \sup_{\mathbf{v}} \sqrt{n}\mathbf{v}^T\mathbf{x} - f^*(\sqrt{n}\mathbf{v}). \tag{28}$$

Translating to our problem and after rescaling this gives,

$$n^{-1/2}f(\mu\mathbf{x}_0 + \mathbf{w}) = \sup_{\mathbf{v}} \mathbf{v}^T(\mu\mathbf{x}_0 + \mathbf{w}) - n^{-1/2}f^*(\sqrt{n}\mathbf{v}). \tag{29}$$

Now, from standard optimality conditions of (28), the optimal $\bar{\mathbf{v}}_*$ satisfies $\bar{\mathbf{v}}_* \in \partial f(\mathbf{x})$. Then, using condition (b) of Section 2.1, $\|\mathbf{v}_*\| = \mathcal{O}(\sqrt{n})$ for all $\mathbf{x}$ such that $\|\mathbf{x}\| = \mathcal{O}(1)$. From this, and $\|\mathbf{w} + \mu \mathbf{x}_0\| = \mathcal{O}(1)$ we conclude that the optimal $\mathbf{v}_*$ in (29) satisfies $\|\mathbf{v}_*\| \leq K_{\mathbf{v}} < 0$ for sufficiently large constant $K_{\mathbf{v}}$ independent of $n$. Putting everything together, (27) is equivalent to

$$\min_{\substack{\|\mathbf{w}\| \leq K_{\mathbf{w}}}} \max_{\substack{\|\mathbf{u}\| \leq 1 \\ 0 \leq \|\mathbf{v}\| \leq K_{\mathbf{v}}}} \frac{1}{\sqrt{n}} \mathbf{u}^T (\mathbf{z_e} - (\mathbf{x}_0^T \mathbf{w})\mathbf{e} - \|\mathbf{P}^\perp \mathbf{w}\|\mathbf{g}) - \|\mathbf{u}\| \bar{\mathbf{h}}^T \mathbf{P}^\perp \mathbf{w}$$

$$+ \lambda \mathbf{v}^T (\mu \mathbf{x}_0 + \mathbf{w}) - \lambda \bar{f}^*(\mathbf{v}), \qquad (30)$$

where we have also denoted $\bar{\mathbf{h}} := n^{-1/2}\mathbf{h}$ and $\bar{f}^*(\mathbf{v}) = n^{-1/2} f^*(\sqrt{n}\mathbf{v})$. Observe again that by condition (b) of Section 2.1, $\bar{f}^*(\mathbf{v}) = \max_{\mathbf{x}} \mathbf{x}^T \mathbf{v} - n^{-1/2} f(\mathbf{x}) = \mathcal{O}(1)$ since $\mathbf{v} = \mathcal{O}(1)$.

In order to somewhat simplify the exposition, we often omit explicitly carrying over the constraints $\|\mathbf{w}\| \leq K_{\mathbf{w}}, \|\mathbf{v}\| \leq K_{\mathbf{v}}$ until the very last step, but we often recall and actually make use of them.

**Optimizing over the directions of u and w.** Observe that the maximization over the direction of $\mathbf{u}$ is easy in (30), which then becomes:

$$\min_{\mathbf{w}} \max_{\substack{0 \leq \beta \leq 1 \\ \mathbf{v}}} \frac{1}{\sqrt{n}} \beta \| \mathbf{z_e} - (\mathbf{x}_0^T \mathbf{w})\mathbf{e} - \|\mathbf{P}^\perp \mathbf{w}\|\mathbf{g} \| - \beta \bar{\mathbf{h}}^T \mathbf{P}^\perp \mathbf{w} + \lambda \mathbf{v}^T (\mu \mathbf{x}_0 + \mathbf{w}) - \lambda \bar{f}^*(\mathbf{v}). \quad (31)$$

At this point, the form of the objective function suggests that it is possible to do the same trick over $\mathbf{w}$, i.e. fix its magnitude and optimize over only its direction. The caveat is that the minimization over $\mathbf{w}$ in (31) is done only after the maximization over $\beta$ and $\mathbf{v}$. What is more, the objective function is not be convex in $\mathbf{w}$; thus, flipping the order of min-max operations that would resolve the issue is not directly justified by (say) Sion's minimax theorem [24].

However, [26] shows that the flipping is possible when dimensions are large. To provide some intuition on why this should be true, observe that the (PO) problem in (24) is itself convex and satisfies all conditions of Sion's minimax theorem; thus the order of min-max operations can be flipped. According to the CGMT, the (PO) and the (AO) are tightly related in an asymptotic setting. [26] uses this to translate the convexity properties of the (PO) to the (AO).[6] Hence, we have:

$$\max_{\substack{0 \leq \beta \leq 1 \\ \mathbf{v}}} \min_{\mathbf{w}} \frac{\beta}{\sqrt{n}} \| \mathbf{z_e} + (\mathbf{x}_0^T \mathbf{w})\mathbf{e} - \|\mathbf{P}^\perp \mathbf{w}\|\mathbf{g} \| - \beta \bar{\mathbf{h}}^T \mathbf{P}^\perp \mathbf{w} + \lambda \mathbf{v}^T (\mu \mathbf{x}_0 + \mathbf{w}) - \lambda \bar{f}^*(\mathbf{v})$$

$$= \max_{\substack{0 \leq \beta \leq 1 \\ \mathbf{v}}} \min_{\alpha_1, \alpha_2 \geq 0} \frac{\beta}{\sqrt{n}} \| \mathbf{z_e} + \alpha_2 \mathbf{e} - \alpha_1 \mathbf{g} \| - \max_{\substack{\|\mathbf{P}^\perp \mathbf{w}\| = \alpha_1 \\ \mathbf{x}_0^T \mathbf{w} = \alpha_2}} \left\{ \beta \bar{\mathbf{h}}^T \mathbf{P}^\perp \mathbf{w} - \lambda \mathbf{v}^T (\mu \mathbf{x}_0 + \mathbf{w}) + \lambda \bar{f}^*(\mathbf{v}) \right\}.$$

By decomposing $\mathbf{w}$ as $\mathbf{P}^\perp \mathbf{w} + (\mathbf{x}_0^T \mathbf{w})\mathbf{x}_0$, it is not hard to perform the maximization over $\mathbf{w}$ to equivalently write the last display above as:

$$\max_{\substack{0 \leq \beta \leq 1 \\ \mathbf{v}}} \min_{\alpha_1, \alpha_2 \geq 0} \frac{\beta}{\sqrt{n}} \| \mathbf{z_e} + \alpha_2 \mathbf{e} - \alpha_1 \mathbf{g} \| - \alpha_1 \|\beta \mathbf{P}^\perp \bar{\mathbf{h}} - \lambda \mathbf{P}^\perp \mathbf{v}\| + \lambda \mu \mathbf{v}^T \mathbf{x}_0 + \alpha_2 \lambda (\mathbf{v}^T \mathbf{x}_0) - \lambda \bar{f}^*(\mathbf{v}).$$

$$(32)$$

**The randomness of e, g and $g$.** Until further notice condition on $\bar{\mathbf{h}}$ and $\mathbf{x}_0$. All randomness in (32) is now on the first term.

Consider $\beta, \mathbf{v}$ fixed for now. For any pair $\alpha_1, \alpha_2$ by the WLLN, $m^{-1}\|\mathbf{z_e} + \alpha_2 \mathbf{e} - \alpha_1 \mathbf{g}\|^2 \xrightarrow{P} \mathbb{E}[(g(\gamma) - \mu\gamma + \alpha_2\gamma - \alpha_1\gamma')^2]$, where $\gamma, \gamma' \sim \mathcal{N}(0,1)$ and independent. Recall, $\mathbb{E}[(g(\gamma) - \mu\gamma)^2] = \sigma^2$, $\mathbb{E}[(g(\gamma) - \mu\gamma)\gamma] = \mu - \mu = 0$ and $m/n = \delta$, to conclude that $n^{-1/2}\|\mathbf{z_e} + \alpha_2 \mathbf{e} - \alpha_1 \mathbf{g}\| \xrightarrow{P} \sqrt{\delta}\sqrt{\sigma^2 + \alpha_1^2 + \alpha_2^2}$, where convergence is point-wise in $\alpha_1, \alpha_2$. Also, the objective function in (32)

$$\max_{\beta \geq 0} \min_{\mathbf{v} \in \mathcal{S}_{\mathbf{v}}} \max_{\substack{\mathbf{u} \in \mathcal{S}_{\mathbf{u}} \\ \|\mathbf{u}\|_2 = \beta}} \|\mathbf{v}\| \mathbf{g}^T \mathbf{u} + \|\mathbf{u}\| \mathbf{h}^T \mathbf{v} + \psi(\mathbf{v}, \mathbf{u}),$$

without changing the conclusions of the CGMT. Please refer to [26] for a proof.

is jointly convex in $[\alpha_1, \alpha_2]$. Thus, point-wise convergence translates to uniform as in [1, Cor.. II.1] , from which, it follows that (for any $\beta, \mathbf{v}$) the minimum over $\alpha_1, \alpha_2$ in (32) converges to

$$\min_{\alpha_1, \alpha_2 \geq 0} \beta \sqrt{\delta} \sqrt{\sigma^2 + \alpha_1^2 + \alpha_2^2} - \alpha_1 \|\beta \mathbf{P}^\perp \bar{\mathbf{h}} - \lambda \mathbf{P}^\perp \mathbf{v}\| + \lambda \mu \mathbf{v}^T \mathbf{x}_0 + \alpha_2 \lambda (\mathbf{v}^T \mathbf{x}_0) - \lambda \bar{f}^*(\mathbf{v}). \quad (33)$$

Furthermore, the function $\sqrt{\sigma^2 + \alpha_1^2 + \alpha_2^2}$ is (by direct differentiation) jointly strongly convex over $[\alpha_1, \alpha_2]$; thus (33) has a unique minimizer. Then, we can apply the Argmin theorem [18, Thm. 2.7] to conclude that the optimal $\alpha_1, \alpha_2$ of (32) converge to the corresponding (unique) optima of (33).

Up to now, $\beta, \mathbf{v}$ were assumed fixed and the convergence from (32) to (33) holds point-wise with respect to $\beta, \mathbf{v}$. The point-wise minimum of concave functions is still concave, thus, uniform convergence is indeed true by [18, Thm. 2.7] . Hence, (32) converges to

$$\max_{\substack{0 \leq \beta \leq 1 \\ \mathbf{v}}} \min_{\alpha_1, \alpha_2 \geq 0} \beta \sqrt{\delta} \sqrt{\sigma^2 + \alpha_1^2 + \alpha_2^2} - \alpha_1 \|\beta \mathbf{P}^\perp \bar{\mathbf{h}} - \lambda \mathbf{P}^\perp \mathbf{v}\| + \lambda \mu \mathbf{v}^T \mathbf{x}_0 + \alpha_2 \lambda (\mathbf{v}^T \mathbf{x}_0) - \lambda \bar{f}^*(\mathbf{v}),$$
$$(34)$$

and the optimal $\alpha_1, \alpha_2$ of the former converge to the corresponding optima of the latter.

**Merging $\alpha_1$ and $\alpha_2$.** It is important to note that $\alpha_1^2 + \alpha_2^2$ in (34) correspond exactly to the squared norm of the error. Here, we simplify (34) by introducing the quantity $\alpha_1^2 + \alpha_2^2$ as the minimization variable rather than sperately $\alpha_1$ and $\alpha_2$. By first order optimality conditions in (34) we find

$$\alpha_1 \beta \sqrt{\delta} = \|\beta \mathbf{P}^\perp \bar{\mathbf{h}} - \lambda \mathbf{P}^\perp \mathbf{v}\| \sqrt{\alpha_1^2 + \alpha_2^2 + \sigma^2} \quad \text{and} \quad -\alpha_2 \beta \sqrt{\delta} = \lambda \mathbf{v}^T \mathbf{x}_0 \sqrt{\alpha_1^2 + \alpha_2^2 + \sigma^2}. \quad (35)$$

Substituting this in (34), the objective becomes (ignoring the terms that do not involve $\alpha_1$ or $\alpha_2$):

$$\beta \sqrt{\delta} \sqrt{\sigma^2 + \alpha_1^2 + \alpha_2^2} - \frac{\sqrt{\sigma^2 + \alpha_1^2 + \alpha_2^2}}{\beta \sqrt{\delta}} \left( \|\beta \mathbf{P}^\perp \bar{\mathbf{h}} - \lambda \mathbf{P}^\perp \mathbf{v}\|^2 + (\lambda \mathbf{v}^T \mathbf{x}_0)^2 \right)$$

But, from (35) we find $\sqrt{\sigma^2 + \alpha_1^2 + \alpha_2^2} \sqrt{\|\beta \mathbf{P}^\perp \bar{\mathbf{h}} - \lambda \mathbf{P}^\perp \mathbf{v}\|^2 + (\lambda \mathbf{v}^T \mathbf{x}_0)^2} = \beta \sqrt{\delta} \sqrt{\alpha_1^2 + \alpha_2^2}$. Combining, we conclude that (34) can be written as

$$\max_{\substack{0 \leq \beta \leq 1 \\ \mathbf{v}}} \min_{\alpha \geq 0} \beta \sqrt{\delta} \sqrt{\sigma^2 + \alpha^2} - \alpha \|\beta \mathbf{P}^\perp \bar{\mathbf{h}} - \lambda \mathbf{v}\| + \lambda \mu \mathbf{v}^T \mathbf{x}_0 - \lambda \bar{f}^*(\mathbf{v}), \quad (36)$$

where the new optimization variable $\alpha$ plays the role of $\sqrt{\alpha_1^2 + \alpha_2^2}$, thus it represents the norm of the error vector $\|\mathbf{w}\|$. We have also identified $\|\beta \mathbf{P}^\perp \bar{\mathbf{h}} - \lambda \mathbf{P}^\perp \mathbf{v}\|^2 + (\lambda \mathbf{v}^T \mathbf{x}_0)^2 = \|\beta \mathbf{P}^\perp \bar{\mathbf{h}} - \lambda \mathbf{v}\|^2$.

**Introducing a new optimization variable**. To get a better handle at it, we square the norm term in (36) at the expense of introducing a new scalar optimization variable. This is based on the following trick:

$$\sqrt{x} = \min_{\tau > 0} \frac{\tau}{2} + \frac{x}{2\tau}, \quad (37)$$

for any $x \geq 0$. Thus, (36) becomes

$$\max_{\substack{0 \leq \beta \leq 1 \\ \mathbf{v}, \tau > 0}} \min_{\alpha \geq 0} \beta \sqrt{\delta} \sqrt{\sigma^2 + \alpha^2} - \frac{\alpha \tau}{2} - \frac{\alpha}{2\tau} \|\beta \mathbf{P}^\perp \bar{\mathbf{h}} - \lambda \mathbf{v}\|^2 + \lambda \mu \mathbf{v}^T \mathbf{x}_0 - \lambda \bar{f}^*(\mathbf{v}), \quad (38)$$

where we have also flipped the order of min-max between $\alpha$ and $\tau$. We could do this as in [22, Cor. 37.3.2] since the objective is convex in $\alpha$ and concave in $\tau$, the constraint sets are both convex and both of them are bounded. To argue the boundedness, recall that $\alpha \leq K_\mathbf{w}$; for $\tau$ it suffices to combine optimality conditions of (37) and boundedness of $\mathbf{v}$, $\|\mathbf{v}\|_2 \leq K_\mathbf{v}$.

**Optimizing over $\mathbf{v}$.** Note that the objective in (38) is concave in $\mathbf{v}$, convex in $\alpha$ and the constraint sets are convex compact. Thus, as it might be expected by now, we use [22, Cor. 37.3.2] to flip the corresponding order of max-min. Also, after some simple algebra while using $\mathbf{P}^\perp \mathbf{x}_0 = 0$ and $\|\mathbf{x}_0\| = 1$, it can be shown that

$$\|\beta \mathbf{P}^\perp \bar{\mathbf{h}} - \lambda \mathbf{v}\|^2 - 2\frac{\tau}{\alpha} \lambda \mu \mathbf{v}^T \mathbf{x}_0 = \|\lambda \mathbf{v} - (\beta \mathbf{P}^\perp \bar{\mathbf{h}} + \frac{\tau}{\alpha} \mu \mathbf{x}_0)\|^2 - \mu^2 \frac{\tau^2}{\alpha^2}.$$

Combining, we conclude with

$$(38) = \max_{\substack{0 \le \beta \le 1 \\ \tau > 0}} \min_{\alpha \ge 0} \beta \sqrt{\delta} \sqrt{\sigma^2 + \alpha^2} - \frac{\alpha \tau}{2} + \mu^2 \frac{\tau}{2\alpha} - \frac{\alpha \lambda^2}{\tau} \min_{\mathbf{v}} \left\{ \frac{1}{2} \|\mathbf{v} - (\frac{\beta}{\lambda} \mathbf{P}^\perp \bar{\mathbf{h}} + \frac{\tau}{\alpha \lambda} \mu \mathbf{x}_0)\|^2 + \frac{\tau}{\lambda \alpha} \bar{f}^*(\mathbf{v}) \right\},$$

(39)

$$= \max_{\substack{0 \le \beta \le 1 \\ \tau > 0}} \min_{\alpha \ge 0} G(\alpha, \beta, \tau).$$

Here, $G(\alpha, \beta, \tau)$ is convex in $\alpha$ (see (36)) and jointly concave in $\beta, \tau$. To see the latter it suffices to show that $\frac{\alpha \lambda^2}{\tau} \|\mathbf{v} - \frac{\beta}{\lambda}(\mathbf{P}^\perp \bar{\mathbf{h}} + \frac{\mu \tau}{\lambda \alpha} \mathbf{x}_0)\|^2$ is jointly convex over $\beta, \tau, \mathbf{v}$ (minimization over $\mathbf{v}$ does not change the joint convexity over $\tau$ and $\beta$.). Norm is separable over its entries, so we equivalently show that for scalars $\tau, \beta, v$, the function $\frac{1}{\tau}(v - c_1 \beta - c_2 \tau)^2$ is jointly convex over $\tau > 0, \beta$; this is true as the perspective function of $(v - c_1 \beta - c_2)^2$. One more remark is in place here regarding the form of $G(\alpha, \beta, \tau)$: even-though $\alpha$ appears in the denominator in (39), the limit of $\alpha \to 0$ of the expression is finite using the continuity of the Moreau envelope [23]. Another way to see this is by noting that the objective in (39) is equivalent to that in (36). Hence, evaluating $G$ at $\alpha = 0$ in the minimization in (39) subsumes $G(0, \beta, \tau) = \lim_{\alpha \to 0} G(\alpha, \beta, \tau)$. (Of course, this is also the case with the optimization problem (10) of Theorem 2.2.)

**The randomness of $\bar{\mathbf{h}}$ and $\mathbf{x}_0$.** Fix $\beta, \tau, \alpha$, denote $c_1 = \frac{\beta}{\lambda}, c_2 = \frac{\tau \mu}{\alpha \lambda}, c_3 = \frac{\tau}{\alpha \lambda}$, and, consider

$$R(\bar{\mathbf{h}}, \mathbf{x}_0) := R(\alpha, \beta, \tau; \bar{\mathbf{h}}, \mathbf{x}_0) := -\frac{c_2}{2} + \frac{1}{c_2} \min_{\mathbf{v}} \left\{ \frac{1}{2} \|\mathbf{v} - c_1 \mathbf{P}^\perp \bar{\mathbf{h}} - c_2 \mathbf{x}_0\|^2 + c_3 \bar{f}^*(\mathbf{v}) \right\}.$$

Recall from Assumption 1, and, from the modeling condition (a) in Section 2.1 that

$$A(\bar{\mathbf{h}}, \mathbf{x}_0) := R(\alpha, \beta, \tau; \bar{\mathbf{h}}, \mathbf{x}_0) := -\frac{c_2}{2} \frac{\|\bar{\mathbf{x}}_0\|^2}{n} + \frac{1}{c_2} \min_{\mathbf{v}} \left\{ \frac{1}{2} \|\mathbf{v} - c_1 \bar{\mathbf{h}} - c_2 \frac{\bar{\mathbf{x}}_0}{\sqrt{n}}\|^2 + c_3 \bar{f}^*(\mathbf{v}) \right\}$$

(40)

converges to $\{-\frac{c_2}{2} + \frac{\mu}{c_2} F(c_1, c_2, c_3)\}$ in probability. Also, recall $\bar{\mathbf{x}}_0 = \mathbf{x}_0 \|\bar{\mathbf{x}}_0\|$. Next, we show that for all constant $\zeta > 0$

$$|R(\bar{\mathbf{h}}, \mathbf{x}_0) - A(\bar{\mathbf{h}}, \mathbf{x}_0)| \le \zeta$$

(41)

with probability approaching one in the limit of $n \to \infty$. Combining this with Assumption 1, will prove that $R(\bar{\mathbf{h}}, \mathbf{x}_0)$ converges in $\{-\frac{c_2}{2} + \frac{\mu}{c_2} F(c_1, c_2, c_3)\}$ in probability, as well.

Proof of (41): Fix any $\epsilon > 0$. We condition on the following events:

$$\begin{cases} |\bar{\mathbf{h}}^T \mathbf{x}_0| \le \epsilon, \\ 1 - \epsilon \le n^{-1/2} \|\bar{\mathbf{x}}_0\| \le 1 + \epsilon. \end{cases}$$

(42)

Each one of the events occurs with probability approaching one as $n \to \infty$; the first follows since $\bar{\mathbf{h}} \sim \mathcal{N}(0, \frac{1}{n} \mathbf{I}_n)$ and $\|\mathbf{x}_0\| = 1$ and from standard tail bounds on Gaussians; the second is due to condition (a) of Section 2.1. Without loss of generality assume $R(\bar{\mathbf{h}}, \mathbf{x}_0) \ge A(\bar{\mathbf{h}}, \mathbf{x}_0)$, and let $\mathbf{v}_*$ be optimal in (40), then

$$|R(\bar{\mathbf{h}}, \mathbf{x}_0) - A(\bar{\mathbf{h}}, \mathbf{x}_0)| \le \frac{c_2}{2} \left( \frac{\|\bar{\mathbf{x}}_0\|^2}{n} - 1 \right) + \frac{1}{2c_2} \|\mathbf{v}_* - c_1 \mathbf{P}^\perp \bar{\mathbf{h}} - c_2 \mathbf{x}_0\|^2 - \frac{1}{2c_2} \|\mathbf{v}_* - c_1 \bar{\mathbf{h}} - c_2 \frac{\bar{\mathbf{x}}_0}{\sqrt{n}}\|^2$$

$$= \frac{c_2}{2} \left( \frac{\|\bar{\mathbf{x}}_0\|^2}{n} - 1 \right)$$

$$+ \left( \frac{c_1}{c_2} (\mathbf{x}_0^T \bar{\mathbf{h}}) \mathbf{x}_0 + \bar{\mathbf{x}}_0 \left( \frac{1}{\sqrt{n}} - \frac{1}{\|\bar{\mathbf{x}}_0\|} \right) \right)^T \left( \mathbf{v}_* - c_1 \bar{\mathbf{h}} - \frac{1}{2} c_2 \bar{\mathbf{x}}_0 \left( \frac{1}{\sqrt{n}} + \frac{1}{\|\bar{\mathbf{x}}_0\|} \right) + \frac{1}{2} c_1 (\mathbf{x}_0^T \bar{\mathbf{h}}) \mathbf{x}_0 \right)$$

$$= -\frac{1}{2} \frac{c_1^2}{c_2} (\mathbf{x}_0^T \bar{\mathbf{h}})^2 + \frac{c_1}{c_2} (\mathbf{x}_0^T \mathbf{h})(\mathbf{x}_0^T \mathbf{v}_*) - c_1 (\mathbf{x}_0^T \bar{\mathbf{h}}) \frac{\|\bar{\mathbf{x}}_0\|}{\sqrt{n}} + (\mathbf{x}_0^T \mathbf{v}_*) \left( \frac{\|\bar{\mathbf{x}}_0\|}{\sqrt{n}} - 1 \right)$$

$$\le \frac{1}{2} \frac{c_1^2}{c_2} \epsilon^2 + \frac{c_1}{c_2} \|\mathbf{v}_*\| \epsilon + c_1 \epsilon (1 + \epsilon) + \|\mathbf{v}_*\| \epsilon$$

(43)

where the last line follows after bounding the absolute values of the summands using (42). Recall now that $\|\mathbf{v}_*\| \leq K_{\mathbf{v}} < \infty$. Also, note that $c_1$ and $\frac{c_1}{c_2}$ are also bounded constants. Then, for all $\zeta > 0$ in (41) we can find sufficiently small $\epsilon > 0$ such that the value of the last expression in the panel above is no larger than $\zeta$, thus completing the proof of (41).

Thus, we have shown that $G(\alpha, \beta, \tau)$ in (39) converges pointwise to

$$H(\alpha, \beta, \tau) := \sqrt{\delta}\sqrt{\sigma^2 + \alpha^2} - \frac{\alpha p}{2} + \mu^2 \frac{\tau}{2\alpha} - \frac{\alpha \lambda^2}{\tau} F(\frac{\beta}{\lambda}, \frac{\tau\mu}{\alpha\lambda}, \frac{\tau}{\alpha\lambda}),$$

in the limit of $n \to \infty$. Note that $H$ is strongly convex in $\alpha$ and jointly concave in $\beta, \mathbf{v}$ since taking limits does not affect convexity properties (recall that $G$ is convex-concave). With these, it follows as per [18, Thm. 2.7] that (i)

$$\max_{0 \leq \beta \leq 1, \tau > 0} \min_{\alpha \geq 0} G(\alpha, \beta, \tau) \xrightarrow{P} \max_{0 \leq \beta \leq 1, \tau > 0} \min_{\alpha \geq 0} H(\alpha, \beta, \tau), \tag{44}$$

and, (ii) $\alpha_*(\mathbf{h}, \mathbf{x}_0) \xrightarrow{P} \alpha_*$, where $\alpha_*$ the unique minimizer of the second optimization in (44). This completes the proof of the theorem.

## A.2 Theorem 2.1

The theorem is a direct consequence of Theorem 2.2. In particular, Theorem 2.2 proves that the value $\alpha_*$ to which the error converges only depends on $g$ through the parameters $\mu$ and $\sigma^2$. Those are the same (by definition) for the non-linear and the linear case considered. Therefore, the errors are the same.

# B Proof of Theorem 2.3

Specializing Theorem 2.2 to the setup of Section 2.2.2 we showed in the same section that $\|\hat{\mathbf{x}} - \mu\mathbf{x}_0\|$ converges in probability to the unique minimizer $\alpha_*$ of the following max-min problem:

$$\max_{\substack{0 \leq \beta \leq 1, \alpha \geq 0 \\ \tau > 0}} \min H(\alpha, \beta, \tau) := \beta\sqrt{\delta}\sqrt{\sigma^2 + \alpha^2} - \frac{\tau\alpha}{2} + \frac{\tau\mu^2}{2\alpha} - \frac{\alpha}{2\tau}\mathbb{E}\left[\eta^2\left(\beta h + \frac{\mu\tau}{\alpha}\overline{X}_0; \lambda\right)\right], \tag{45}$$

where the expectation is over $h \sim \mathcal{N}(0,1)$ and $\overline{X}_0 \sim p_{\overline{X}_0}$. Here, we prove Theorem 2.3 by analyzing the optimality conditions of (45). Recall as in the proof of Theorem 2.2 that $H$ is jointly concave in $\beta, p$ and strongly convex in $\alpha$.

## B.1 First Order Optimality Conditions.

We begin with a lemma, which characterizes the first-order optimality conditions of (45).

**Lemma B.1** (Optimality Conditions). *Consider the following pair of equations with respect to $\beta$ and $\kappa$:*

$$\begin{cases} \beta^2\kappa^2\delta = \sigma^2 + \mathbb{E}\left[(\eta(\beta\kappa h + \mu\overline{X}_0; \kappa\lambda) - \mu\overline{X}_0)^2\right], & (46) \\ \beta\kappa\delta = \mathbb{E}[(\eta(\beta\kappa h + \mu\overline{X}_0; \kappa\lambda) \cdot h)]. & (47) \end{cases}$$

*Also, define $\lambda_{min}$ to be the unique non-negative solution to the equation*

$$(1 + x^2)\int_{-\infty}^{-x} e^{-z^2/2}\mathrm{d}z - xe^{-x^2/2} = \delta\sqrt{\frac{\pi}{2}}.$$

*With these, let $(\beta_*, \tau_*, \alpha_*)$ be optimal in (45). Then,*

$$\alpha_*^2 = \beta_*^2\kappa_*^2\delta - \sigma^2 \quad and \quad \kappa_* = \frac{\sigma}{\sqrt{\beta_*^2\delta - \tau_*^2}}. \tag{48}$$

*such that,*

(i) *If $\beta_* = 1$ and $\lambda > \lambda_{min}$, then $\kappa_*$ is the unique solution to (46) for $\beta = 1$,*

*(ii) If $\beta_* \in (0,1)$ ,then $\kappa_*, \beta_*$ are solutions to the pair of equation (46)-(47).*

*Proof.* Consider the derivation of the objective function with respect to $\alpha$, $\tau$ and $\beta$ as follows

$$\frac{\partial}{\partial \alpha} H(\alpha, \tau, \beta) = \frac{\beta \alpha \sqrt{\delta}}{\sqrt{\alpha^2 + \sigma^2}} - \frac{\tau}{2} - \frac{1}{2\tau} \mathbb{E}[(\eta(\beta h + \frac{\mu \tau}{\alpha} \overline{X}_0, \lambda) - \frac{\mu \tau}{\alpha} \overline{X}_0)^2] \tag{49}$$

$$\frac{\partial}{\partial \tau} H(\alpha, \tau, \beta) = -\frac{\alpha}{2} + \frac{\alpha}{2\tau^2} \mathbb{E}[(\eta(\beta h + \frac{\mu \tau}{\alpha} \overline{X}_0, \lambda) - \frac{\mu \tau}{\alpha} \overline{X}_0)^2] \tag{50}$$

$$\frac{\partial}{\partial \beta} H(\alpha, \tau, \beta) = \sqrt{\delta} \sqrt{\alpha^2 + \sigma^2} - \frac{\alpha}{\tau} \mathbb{E}[\eta(\beta h + \frac{\mu \tau}{\alpha} \overline{X}_0, \lambda) \cdot h] \tag{51}$$

We will prove that the optimal $\alpha_*, \tau_*$ and $\beta_*$ are all strictly positive. First, suppose $\alpha_* = 0$ and $\tau_* > 0$. Then, the first term in (49) goes to zero while $-\frac{\tau}{2}$ stays negative and the final term is always non-positive. This shows $\frac{\partial}{\partial \alpha}\big|_{\alpha \to 0} H(\alpha, \tau, \beta) < 0$, which means that $\alpha_* = 0$ cannot be optimal in this case by convexity. Next, assume $\alpha_* = 0$ and the optimal over $\tau$ is approached as $\tau \to 0$. In this case, it can be shown that the expected value in (49) is strictly positive, thus the derivative remains negative. Thus, $\alpha_* > 0$, as promised. A similar argument shows strict positivity of (50) when $\tau \to 0$. Thus, $\tau_* > 0$. Finally, $\frac{\partial}{\partial \beta}\big|_{\beta \to 0} H(\alpha, \tau, \beta) > 0$, by independence of $h$ and $\overline{X}_0$, showing $\beta_* > 0$.

The argument above shows that the derivatives are equal to zero at the optimal. For convenience define

$$P\left(\frac{\tau}{\alpha}\right) := \frac{\tau \mu^2}{2\alpha} - \frac{\alpha \lambda^2}{2\tau} \mathbb{E}\left[\eta^2 \left(\frac{\beta}{\lambda} h + \frac{\mu \tau}{\lambda \alpha} \overline{X}_0; 1\right)\right].$$

Then equating the derivatives in (45) with respect to $\alpha$ and $\tau$ with zero gives

$$\frac{\beta \sqrt{\delta} \alpha}{\sqrt{\alpha^2 + \sigma^2}} - \frac{\tau}{2} - \frac{\tau}{\alpha^2} P'(\frac{\tau}{\alpha}) = 0, \tag{52a}$$

$$-\frac{\alpha}{2} + \frac{1}{\alpha} P'(\frac{\tau}{\alpha}) = 0. \tag{52b}$$

Here, $P'$ is the derivative of $P(x)$ with respect to $x$. Any optimal $\beta_*, \tau_*, \alpha_*$ satisfies these. Then, it only takes multiplying (52b) by $\frac{\tau}{\alpha}$ and adding the result to (52a) to see that

$$\alpha_* = \frac{\tau_* \sigma}{\sqrt{\beta^2 \delta - \tau_*^2}}. \tag{53}$$

Next, substituting (53) in (52b) it can be shown that,

$$-\frac{\sigma^2}{2} + \frac{\sigma^2}{2\tau^2} \mathbb{E}[(\eta(\beta h + \frac{\sqrt{\beta^2 \delta - \tau^2}}{\sigma} \mu \overline{X}_0; \lambda) - \frac{\sqrt{\beta^2 \delta - \tau^2}}{\sigma} \mu \overline{X}_0)^2] = 0.$$

To reach this we have also used the following facts: $\eta(x; \lambda) \frac{\partial}{\partial x} \eta(x; \lambda) = \eta(x; \lambda)$, $\lambda \eta(\frac{x}{\lambda}; 1) = \eta(x; \lambda)$ and $\mathbb{E}[\overline{X}_0^2] = 1$ by assumption. Multiplying the result with $2\tau^2/\sigma^2$ and defining

$$\kappa := \frac{\sigma}{\sqrt{\beta^2 \delta - \tau^2}},$$

we conclude with,

$$\beta^2 \delta \kappa^2 - \sigma^2 = \mathbb{E}[(\eta(\beta \kappa h + \mu \overline{X}_0; \kappa \lambda) - \mu \overline{X}_0)^2], \tag{54}$$

which is same as (46). Also, with respect to the optimal $\kappa_*$ it is easily seen by (53) that

$$\alpha_*^2 = \beta_*^2 \kappa_*^2 \delta - \sigma^2. \tag{55}$$

The derivative in (45) with respect to $\beta$ gives

$$\frac{\partial}{\partial \beta} H(\alpha, \beta, \tau) = \sqrt{\delta} \sqrt{\sigma^2 + \alpha^2} - \frac{\alpha}{\tau} \mathbb{E}[\eta(\beta h + \frac{\mu \tau}{\alpha} \overline{X}_0; \lambda) h]$$

$$= \beta \delta \kappa - \kappa \mathbb{E}[\eta(\beta h + \frac{\mu \overline{X}_0}{\kappa}, \lambda) h] = \beta \delta \kappa - \mathbb{E}[\eta(\kappa \beta h + \mu \overline{X}_0; \lambda \kappa) h]. \tag{56}$$

where we have also used (55). Note that the above is same as (47) and recall the constraint $0 \le \beta \le 1$ in (45) to conclude with the desired.

It only remains to show that the solution with respect to $\kappa$ of (46) (eqv. of (54)) is unique when $\beta = 1$ and $\lambda \ge \lambda_{\min}$. For $\beta = 1$, (46) is the same as fixed point equation [3, Eqn. (1.9)], which in turn was shown to admit a unique solution for all $\lambda > \lambda_{\min}$ in [11] (see [3, Prop. 1.3]). $\qquad \square$

## B.2 The Regions of Operation

We build up to the proof of Theorem 2.3 through a series of auxiliary lemmas. Through the lemmas, we identify two "regimes of operation" of the LASSO. The first, we call $\mathcal{R}_{\text{bad}}$, and it corresponds to values of $\lambda$ for which the optimal $\beta$ is in the open set $(0, 1)$. The second regime, is such that $\beta = 1$. If $\delta < 1$, we prove in Lemma B.5 that there exists a unique critical value $\lambda_{\text{crit}}$ separating the two regimes in the sense that $\mathcal{R}_{\text{bad}}$ extends from 0 to $\lambda_{\text{crit}}$. If on the other hand $\delta \ge 1$, then there is no $\mathcal{R}_{\text{bad}}$ region (Lemma B.6).

First, we need a few useful definitions.

**Definition B.1.** *For any $\lambda > 0$, we let $\alpha_*(\lambda)$, $\tau_*(\lambda)$ and $\beta_*(\lambda)$ be optimal solutions in (45). Apart from $\alpha_*(\lambda)$, the others are not necessarily unique at this point. Also, $\kappa_*(\lambda)$ is defined as in (48).*

**Definition B.2** (Bad Regime). *We say that a value $\lambda > 0$ is in the bad regime $\mathcal{R}_{bad}$, denote $\lambda \in \mathcal{R}_{bad}$, if there exists $\beta_*(\lambda) \in (0, 1)$.*

**Definition B.3** (Critical Regime). *We say that a value $\lambda_{crit} > 0$ is in the critical regime $\mathcal{R}_{crit}$, denote $\lambda_{crit} \in \mathcal{R}_{crit}$ if for some $\kappa_{crit}$, the pair $\lambda_{crit}, \kappa_{crit}$ solves:*

$$\begin{cases} \kappa^2 \delta = \sigma^2 + \mathbb{E}\left[ (\eta(\kappa h + \mu \overline{X}_0; \kappa \lambda) - \mu \overline{X}_0)^2 \right], & (57) \\ \kappa \delta = \mathbb{E}[(\eta(\kappa h + \mu \overline{X}_0; \kappa \lambda) \cdot h)]. & (58) \end{cases}$$

As an immediate consequence of the definition above and the first order optimality conditions in Lemma B.1, we have

$$\beta_*(\lambda_{\text{crit}}) = 1, \quad \kappa_*(\lambda_{\text{crit}}) = \kappa_{\text{crit}} \quad \text{and} \quad \alpha_*(\lambda_{\text{crit}}) = \sqrt{\delta \kappa_{\text{crit}}^2 - \sigma^2}. \tag{59}$$

Also, the following lemma reveals the importance of $\lambda_{\text{crit}}$: all $\lambda < \lambda_{\text{crit}}$ are in $\mathcal{R}_{\text{bad}}$ and the squared error is constant in that regime, i.e. $\alpha_*(\lambda) = \alpha_*(\lambda_{\text{crit}})$.

**Lemma B.2** (Error in $\mathcal{R}_{bad}$). *Let $\lambda_{crit} \in \mathcal{R}_{crit}$. Then, for all $0 < \lambda' < \lambda_{crit}$, it holds $\lambda' \in \mathcal{R}_{bad}$. Furthermore, $\beta_*(\lambda') = \lambda/\lambda_{crit}$, $\lambda' \kappa_*(\lambda') = \kappa_{crit} \lambda_{crit}$ and $\alpha_*(\lambda') = \alpha_*(\lambda_{crit})$.*

*Proof.* Fix any $0 < \lambda' < \lambda_{\text{crit}}$. By definition, there exists $\kappa_{\text{crit}}$ such that $\lambda_{\text{crit}}, \kappa_{\text{crit}}$ satisfy (57)-(58). Define $\beta' := \lambda/\lambda_{\text{crit}}$ and $\kappa' := \kappa_{\text{crit}}/\beta'$. It is then easy to see that $\beta', \kappa'$ solve (46)-(47) (for $\lambda = \lambda'$ therein). Also, $\beta' < 1$ by definition. Thus, $\lambda' \in \mathcal{R}_{\text{bad}}$ and $\beta_*(\lambda') = \lambda/\lambda_{\text{crit}}$, $\kappa_*(\lambda') = \kappa_{\text{crit}} \lambda_{\text{crit}}/\lambda'$. Also, using (48) and (59), $\alpha_*(\lambda) = \sqrt{\delta \beta_*^2(\lambda') \kappa_*^2(\lambda') - \sigma^2} = \sqrt{\delta \kappa_*^2(\lambda_{\text{crit}}) - \sigma^2} = \alpha_*(\lambda_{\text{crit}})$. $\qquad \square$

It is thus important to identify the critical values of the regularizer parameter, i.e. all $\lambda_{\text{crit}} \in \mathcal{R}_{\text{crit}}$. Values in $\mathcal{R}_{\text{bad}}$ are important towards this direction, since as shown in the next lemma, for any $\lambda \in \mathcal{R}_{\text{bad}}$ there must exist some $\lambda_{\text{crit}} > \lambda$.

**Lemma B.3** ($\mathcal{R}_{bad} \to \lambda_{crit}$). *Let $\lambda_1 \in \mathcal{R}_{bad}$, then there exists $\lambda_2 \in \mathcal{R}_{crit}$ with $\lambda_2 > \lambda_1$.*

*Proof.* Let $\beta_1, \alpha_1, \kappa_1$ be optimal corresponding to $\lambda_1$. Since $\lambda_1 \in \mathcal{R}_{\text{bad}}$, it holds $0 < \beta_1 < 1$. Then, from Lemma B.1, $\kappa_1, \beta_1$ solve (46)-(47). Starting from these and substituting $\lambda_2 := \lambda_1/\beta_1$ and $\kappa_2 := \kappa_1 \beta_1$ therein, it is not hard to see that this is equivalent with $\lambda_2, \kappa_2$ satisfying (57)-(58). Thus, $\lambda_2 \in \mathcal{R}_{\text{crit}}$. Also, clearly $\lambda_2 > \lambda_1$. $\qquad \square$

The lemma below is important since it shows that when $\delta < 1$ there exists a *unique* $\lambda_{\text{crit}} \in \mathcal{R}_{\text{crit}}$.

**Lemma B.4** (Unique $\lambda_{crit}$). *Suppose $\delta < 1$. The set of equations (57)-(58) has a* unique *pair of solutions $(\kappa, \lambda)$. Thus, there exists unique $\lambda_{crit} \in \mathcal{R}_{crit}$.*

*Proof.* First, we show that there exists at most one $\lambda_{\text{crit}} \in \mathcal{R}_{\text{crit}}$. For the shake of contradiction assume two different pairs of solutions, say $(\kappa_1, \lambda_1)$ and $(\kappa_2, \lambda_2)$. By definition, $\lambda_1, \lambda_2 \in \mathcal{R}_{\text{crit}}$. First, note that we cannot have $\lambda_1 = \lambda_2$, since if this was the case then from (59) we would also have $\kappa_1 = \kappa_2$. Henceforth, assume w.l.o.g. that $\lambda_1 < \lambda_2$. It follows from Lemma B.2 that $\lambda_1 \in \mathcal{R}_{\text{bad}}$ and also $\kappa_*(\lambda_1)\lambda_1 = \kappa_*(\lambda_2)\lambda_2$. Thus,

$$\kappa_*(\lambda_1) < \kappa_*(\lambda_2). \tag{60}$$

But also, again from Lemma B.2, $\alpha_*(\lambda_1) = \alpha_*(\lambda_2)$. Since, $\lambda_1, \lambda_2 \in \mathcal{R}_{\text{crit}}$, this implies when combined with (59) that $\kappa_*(\lambda_1) = \kappa_*(\lambda_2)$, which contradicts (60), completing the proof of this part.

Let us now prove that $\mathcal{R}_{\text{crit}}$ is non-empty. To begin with, we show that $\mathcal{R}_{\text{bad}}$ is non-empty in this case. In particular, we show that $\lambda_{\min}$ defined in Lemma B.1 is in $\mathcal{R}_{\text{bad}}$. Since, $\delta < 1$, we have $\lambda_{\min} > 0$. Suppose that $(\beta_*(\lambda_{\min}) = 1, \kappa_*(\lambda_{\min}))$ is optimal for some $\kappa_*(\lambda_{\min})$, then, from first-order optimality conditions, $\kappa_*(\lambda_{\min}), \lambda_{\min}$ solves (46) for $\beta = 1$. But, then as in [3, pg. 16] $\kappa_*(\lambda_{\min}) \to \infty$. Also, since $H(\alpha, \tau, \beta)$ is concave in $\beta$, the above imply that $\frac{\partial H}{\partial \beta}\big|_{(\beta=0, \kappa \to \infty)} \geq 0$, or equivalently from (56),

$$\int_{\lambda_{\min}}^{\infty} h(h - \lambda_{\min})e^{-h^2/2}dh \leq \delta\sqrt{\frac{\pi}{2}}.$$

Recalling the definition of $\lambda_{\min}$ in Lemma B.1, it can be shown (using standard inequalities on tail functions of gaussians) that the inequality above is violated for all $0 < \delta < 1$. Hence, it must be $\beta_*(\lambda_{\min}) < 1$. Also, $\beta_*(\lambda_{\min}) > 0$ because of (53). Thus, $\lambda_{\min} \in \mathcal{R}_{\text{bad}}$. To complete, the proof use Lemma B.3 with $\lambda_1 = \lambda_{\min}$ to see that there exists $\lambda_2 \in \mathcal{R}_{\text{crit}}$. $\qquad\square$

**Lemma B.5** ($\delta < 1$). *Suppose $\delta < 1$ and let $\lambda_{crit} \in \mathcal{R}_{crit}$. Furthrermore, i) for all $\lambda \leq \lambda_{crit}$, $\alpha_*(\lambda) = \alpha_*(\lambda_{crit})$, and, ii) for all $\lambda > \lambda_{crit}$, $\kappa_*(\lambda)$ is the unique solution to (46) for $\beta = 1$.*

*Proof.* Existence and uniqueness of $\lambda_{\text{crit}}$ is proved in Lemma B.4

i) For $\lambda \leq \lambda_{\text{crit}}$, the claim follows directly from Lemma B.2.

ii) Next, we show that for $\lambda \geq \lambda_{\text{crit}}$, there exists an optimal solution for which $\beta_*(\lambda) = 1$. This suffices since then $\kappa_*(\lambda)$ solves (46) for $\beta = 1$ (by first order optimality conditions), and, also, the solution is unique by [11], [3, Prop. 1.3] and the fact that $\lambda_{\min} \leq \lambda_{\text{crit}} \leq \lambda$. To see that $\beta_*(\lambda) = 1$, we argue as follows. First, $\beta_*(\lambda) \notin (0, 1)$. Otherwise, $\lambda \in \mathcal{R}_{\text{bad}}$, thus, by Lemma B.3 there exists $\lambda' > \lambda \geq \lambda_{\text{crit}}$ such that $\lambda' \in \mathcal{R}_{\text{crit}}$, which contradicts the uniqueness of $\lambda_{\text{crit}}$. Hence, $\beta_*(\lambda) = 1$. $\quad\square$

**Lemma B.6** ($\delta > 1$). *Suppose $\delta > 1$, then for all $\lambda > 0$, $\kappa_*(\lambda)$ is the unique solution to (46) for $\beta = 1$.*

*Proof.* First, let us show that for $\lambda \to 0$, $\beta_*(\lambda) = 1$. Indeed for $\beta = 1$ and $\lambda \to 0$, (56) gives

$$\frac{\partial H}{\partial \beta} = \delta - \mathbb{E}[(h + \frac{\mu}{\kappa}\overline{X}_0)h] = \delta - 1 > 0.$$

Thus, from concavity of $H$ with respect to $\beta$, we find that the unique optimal value for $\beta$ is

$$\beta_*(\lambda \to 0) = 1. \tag{61}$$

Also, as in the proof of Lemma B.5, $\beta_*(\lambda \to \infty) = 1$. Thus, again similar to Lemma B.5, it suffices to prove that there exists no $\lambda \in \mathcal{R}_{\text{bad}}$. For the shake of contradiction, suppose that there exists $\lambda_1 \in \mathcal{R}_{\text{bad}}$. By Lemma B.3, there exists $\lambda_1 < \lambda_{\text{crit}} \in \mathcal{R}_{\text{crit}}$. But, then $\beta_*(\lambda \to 0) \to 0$, which contradicts (61). This completes the proof. $\quad\square$

*Proof.* (of Theorem 2.3) The claim of the theorem is now a direct consequence of Lemmas B.5 and B.6 combined with (55). $\quad\square$

## C Proofs for Section 2.3

### C.1 The LM Algorithm

The Lloyd-Max algorithm is an algorithm for finding the quantization threshold $t_i$ and the representation points $\ell_i$. Given real values $x \in \mathbb{R}$ sampled from some probability density $\phi(x)$ it looks for optimal sets $\hat{\mathbf{t}}, \hat{\boldsymbol{\ell}}$ that minimizes the mean-square-error (MSE) between $x$ and their corresponding quantized values $Q_q(x; \boldsymbol{\ell}, \mathbf{t})$, i.e.

$$(\hat{\boldsymbol{\ell}}, \hat{\mathbf{t}}) := \arg\min_{\boldsymbol{\ell}, \mathbf{t}} \mathbb{E}_{x \sim \phi}[(x - Q_q(x; \boldsymbol{\ell}, \mathbf{t}))^2]. \tag{62}$$

The algorithm simply alternates between i) optimizing the threshold $\mathbf{t}_i$ for a given set of $\boldsymbol{\ell}$, and then ii) optimizing the levels $\boldsymbol{\ell}_i$ for the new thresholds. It is well known that the converging points $\boldsymbol{\ell}^{LM}, \mathbf{t}^{LM}$ of the algorithm satisfy

$$\mathbf{t}_i^{LM} = \frac{\boldsymbol{\ell}_i^{LM} + \boldsymbol{\ell}_{i+1}^{LM}}{2} \qquad\qquad i = 1, ..., L-1, \tag{63a}$$

$$\boldsymbol{\ell}_i^{LM} = \left( \int_{\mathbf{t}_{i-1}^{LM}}^{\mathbf{t}_i^{LM}} \phi(x)\mathrm{d}x \right)^{-1} \left( \int_{\mathbf{t}_{i-1}^{LM}}^{\mathbf{t}_i^{LM}} x\phi(x)\mathrm{d}x \right) \qquad i = 1, ..., L. \tag{63b}$$

Furthermore, they are stationary points of the objective function in (62).

#### C.1.1 Gaussian case

Assume that the values $x$ are sampled from a standard gaussian distribution, i.e. $x \sim \mathcal{N}(0, 1)$ and $\phi(x) = (1/\sqrt{2\pi}) \exp(-x^2/2)$. Also, recall the definition of the parameters $\mu, \sigma^2$ in (3); setting $g = Q_q$ therein, we find (also, to compare with (20))

$$\mu := \mu(\boldsymbol{\ell}, \mathbf{t}) = 2 \sum_{i=1}^{L} \ell_i \int_{\mathbf{t}_{i-1}}^{\mathbf{t}_i} x\phi(x)\mathrm{d}x, \tag{64a}$$

$$\tau^2 := \tau^2(\boldsymbol{\ell}, \mathbf{t}) = 2 \sum_{i=1}^{L} \ell_i^2 \int_{\mathbf{t}_{i-1}}^{\mathbf{t}_i} \phi(x)\mathrm{d}x, \tag{64b}$$

In this notation, the objective in (62) can be written as $\tau^2 - 2\mu + 1$. Thus, $\boldsymbol{\ell}^{LM}, \mathbf{t}^{LM}$ satisfy

$$(\tau^2)'\big|_{(\boldsymbol{\ell}^{LM}, \mathbf{t}^{LM})} = 2\mu'\big|_{(\boldsymbol{\ell}^{LM}, \mathbf{t}^{LM})}, \tag{65}$$

Here and onwards we use $(\tau^2)'$, $\mu'$ to denote the gradient of $\tau^2$ and $\mu$ with respect to the vector $[\boldsymbol{\ell}^T, \mathbf{t}^T]$. The gradients are evaluated at the point $(\boldsymbol{\ell}^{LM}, \mathbf{t}^{LM})$ in (65).

### C.2 q-Bit Compressive Sensing

We prove that the LM algorithm is an efficient algorithm when the objective is minimizing the LASSO reconstruction error of a signal $\mathbf{x}_0$ to which we have access through $q$-bit quantized linear measuments $Q_q(\mathbf{a}_i^T \mathbf{x}; \boldsymbol{\ell}, \mathbf{t})$. It was shown in Section 2.3.2 that the problem can be posed as that of finding $\boldsymbol{\ell}_*, \mathbf{t}_*$ such that

$$(\mathbf{t}_*, \boldsymbol{\ell}_*) = \arg\min_{\mathbf{t}, \boldsymbol{\ell}} \frac{\sigma^2(\mathbf{t}, \boldsymbol{\ell})}{\mu^2(\mathbf{t}, \boldsymbol{\ell})} = \arg\min_{\mathbf{t}, \boldsymbol{\ell}} \frac{\tau^2(\mathbf{t}, \boldsymbol{\ell})}{\mu^2(\mathbf{t}, \boldsymbol{\ell})}. \tag{66}$$

The following Lemma proves the claim made in Section 2.3.3, i.e. the converging points of the LM algorithm are stationary points of the objective function in (66).

**Lemma C.1.** *The converging points of the LM algorithm, say $(\mathbf{t}^{LM}, \boldsymbol{\ell}^{LM})$ satisfy*

$$\frac{\partial}{\partial \ell_i} \left( \frac{\tau^2(\boldsymbol{\ell}, \mathbf{t})}{\mu^2(\boldsymbol{\ell}, \mathbf{t})} \right) \Bigg|_{(\boldsymbol{\ell}, \mathbf{t}) = (\boldsymbol{\ell}^{LM}, \mathbf{t}^{LM})} = 0 \qquad\qquad , i = 1, ..., L,$$

$$\frac{\partial}{\partial t_i} \left( \frac{\tau^2(\boldsymbol{\ell}, \mathbf{t})}{\mu^2(\boldsymbol{\ell}, \mathbf{t})} \right) \Bigg|_{(\boldsymbol{\ell}, \mathbf{t}) = (\boldsymbol{\ell}^{LM}, \mathbf{t}^{LM})} = 0 \qquad\qquad , i = 0, ..., L-1. \tag{67}$$

*Proof.* Call $R(\mathbf{t}, \boldsymbol{\ell}) = \frac{\tau^2(\mathbf{t}, \boldsymbol{\ell})}{\mu^2(\mathbf{t}, \boldsymbol{\ell})}$. We denote $R' := R'(\mathbf{t}, \boldsymbol{\ell})$ for its gradient with respect to the vector $[\mathbf{t}^T, \boldsymbol{\ell}^T]$. It suffices to prove that $R'\big|_{(\boldsymbol{\ell}^{LM}, \mathbf{t}^{LM})} = 0$, or equivalently, that at the point $(\mathbf{t}, \boldsymbol{\ell}) = (\mathbf{t}^{LM}, \boldsymbol{\ell}^{LM})$ the following holds:

$$(\tau^2)' \mu^2 = 2\tau^2 \mu \mu'. \tag{68}$$

To see that this is the case, note that

$$\tau^2(\mathbf{t}^{LM}, \boldsymbol{\ell}^{LM}) = \mu(\mathbf{t}^{LM}, \boldsymbol{\ell}^{LM}) \tag{69}$$

This follows by direct substitution of (63) in (64). Then, (68) follows from (69) and (65). $\qquad\square$