[Reviews · NeurIPS 2015]

Submitted by Assigned_Reviewer_1

Answer to authors' response:

The reviews of other reviewers and the authors' response have not significantly affected my opinion of the paper. That said, I am not familiar with the single index model/sufficient dimension reduction literature, so I could very well be unaware of previous work that may be relevant.
Summary: The authors derive asymptotic guarantees for the generalized square-root lasso applied to Gaussian measurements that are (optionally) transformed by a possibly unknown nonlinear function. The results are very interesting, as they provide a precise characterization of the performance of the method, and the paper is very well written (although there are quite a few typos. especially in the proof section).

Submitted by Assigned_Reviewer_2

The article extends the work on precisely characterizing the statistical performance of the l2-lasso to single-index models. The article also relaxes the usual Gaussian assumption on the prior (the distribution of x_0). It is a worthwhile contribution to the literature on precise asymptotic behavior of lasso-type estimators.

The article is well-written. However,

1. as pointed out by Brillinger (1982), the single index model is equivalent to a linear model with scaled unknown signal and non-standard noise. The connection is desribed rigorously in Appendix A, but it should also be mentioned in the main article to heuristically justify the statement in the blockquote on line 140.

2. the article focuses on the l2-lasso, also called the square root-lasso. The main benefit of the l2-lasso over the popular l2^2-lasso is the ``oracle'' value of the regularization parameter has no dependence on the (usually unknown) noise variance (sigma). A practical question that the results could shed light on is whether this justification for the l2-lasso over the l2^2-lasso is valid.

Summary: The article extends the work on precisely characterizing the statistical performance of the l2-lasso to single-index models. It is a worthwhile contribution to the literature on precise asymptotic behavior of lasso-type estimators.

Submitted by Assigned_Reviewer_3

Summary: This paper considers the Generalized Lasso with non-linear measurements and Gaussian design. A known heuristic often used is that non-linear observations may be treated as noisy linear observations. Capitalizing on the work of Plan and Vershynin [17], the authors here extend the results of [17], and by moving to an asymptotic regime, they provide novel precise explicit expressions of the error estimation. The accuracy of their predictions is confirmed with many special cases (including the Lasso, the group Lasso and the nuclear norm). The application to the design of optimal quantizers in the context of q-Bit CS and its relation to the Lloyd-Max quantizer is very insightful.

Quality: This is a nice piece of work.

Clarity: The paper is quite well-written, though the treatment is somewhat unbalanced. The lack of space however prevents including all the necessary details.

Originality: The paper builds upon the work of Plan and Vershynin [17] and expands in several aspects that are novel. For instance, the explicit expressions of the performance estimation error are new.

Comments:

1) Page 4, line 193: the regularized and penalized versions are equivalent with appropriate correspondence.

2) Caption of Figure 1: 'n=768' shouldbe '$n$=768'.

3) Page 5, line 247: a typo at 'When ...'.

4) Page 5, line 294: what the authors call the proximal function is (very well) known in convex analysis as the Moreau envelope of index \tau. The notation used there is awkward as prox is generally used for the proximal point/operator but not for the Moreau envelope. I suggest changing it.

5) Page 5, Assumption 1: I think it would instructive to give this assumption a more formal interpretation using results from variational analysis.

6) The authors should avoid calling the generalized Lasso an algorithm in the text. It is certainly a (convex) program (or decoder to borrow the terminology widely used in CS), but not an algorithm.

7) Experiments: the authors show the results when applying the generalized Lasso to the non-linear measurements, and how the theorems predict the estimation error.
Summary: This paper considers the Generalized Lasso with non-linear measurements and Gaussian design are proves that asymptotically, the estimation performance of the Generalized Lasso with such measurements is the same as if the measurements were linear and noisy (Gaussian). The authors provide novel precise explicit expressions of the estimation error, and confirm their findings with many special cases (including the Lasso, the group Lasso and the nuclear norm). The application to the design of optimal quantizers in the context of q-Bit CS and its relation to the Lloyd-Max quantizer is very insightful.

Submitted by Assigned_Reviewer_4

This paper considers about linear estimation with nonlinear link function. Inspired by the recent work of Plan and Vershynin, it studies the asymptotical estimation error using generalized lasso for arbitrary link function with \mu \ne 0. It turns out that no matter the link function is linear or nonlinear, the asymptotical error is exactly the same as shown in theorem 2.1.

Theorem 2.1 holds when the linear measurement vector is sampled from normal distribution, which might be strict in some cases. The author might need to explain why this assumption is necessary for the conclusion to hold.

A few questions: 1. How to choose regularizer based on set \kai in which x_0 lies? While the assumption of convex regularizer is discussed in this paper, it seems the aforementioned problem is not addressed yet. 2. While the result shows nonlinear is equal to linear measurement, but the conclusion involves with \mu and \sigma which characterizes the hardness of estimation from nonlinear function g. I would like to know if the authors have any thought on the necessity of these dependences.

There are some references missing. 1. Single index model and sufficient dimension reduction study linear vector/subspace estimation under unknown link function. However, the paper does not review any paper in this area. 2. A recent paper "Optimal linear estimation under unknown linear estimation" by Yi et al. introduces new algorithm for sparse recovery even with link function that has zero \mu.
Summary: The paper establishes the equivalence of high dimensional linear estimation with linear and nonlinear link function under proper conditions. The conclusion is supported by careful theoretical proof and empirical results. This is a well written paper with fairly interesting results.

Author Feedback
Author rebuttal: We thank all the reviewers for their comments.

Rev_1
1) Please allow us to clarify the distinction between our result, as summarized on line 140, and that of [5]. Indeed, Brillinger was the first to identify the connection between the single-index model and a properly defined linear one. His result applies to the OLS estimate and only to the typical setting of interest at the time where m grows large, while n stays fixed. Our result shows that the connection identified by Brillinger holds in a *much more general* setting; we consider the Generalized LASSO (for which OLS is only a very special case), and the regime where both m and n can grow large. We believe that this discussion on the relevance of our result to the classical one by Brillinger as detailed in Sec. 1.5.1, serves itself as an initial justification of our findings.

2) Indeed, we believe that our result can justify the fact that the tuning of the l2 lasso achieving "oracle" performance is independent of \sigma. Earlier work [18] on the precise analysis of the l2-lasso with linear measurements has verified this. Owing to the established equivalence of the lasso with single-index model to that with linear measurements, we can translate the same conclusion to the single-index model.

Rev_2
We thank the reviewer for the encouraging feedback. Raised points #1-4&6 will be addressed in the revised manuscript. Regarding point #5, it is unclear to us how to do this, as we do not understand what the reviewer means by "variational analysis".

Rev_3
Let us begin with a comment on the Gaussianity assumption. It plays a critical role in our proof: i) in the decomposition (line 572), ii) in applying the CGMT theorem. We note that empirical results have shown that, in the case of the lasso, the CGMT might still be applicable for iid measurements from a wider class of distributions[24]. The Gaussian decomposition property, also used by Brillinger and Plan&Vershynin, appears to be stricter. However, some aspects of Brillinger's result have been shown to hold with relaxed conditions on the measurements (e.g. Duan&Li (1991)); it might be an interesting direction for future work to explore such extensions under the setting of our paper.

On the two questions:
1) Our focus is entirely on the regularized version of the lasso. We only refer to the constrained lasso(Sec. 1.5.1), in which the set \kai appears, so as to connect our result to that of [19], where only this later version is analyzed. Guidelines on choosing the regularizer according to prior knowledge on x0 have been studied in the literature (e.g. [6], Sec. 1.2).

2) Thm. 2.2 shows that the asymptotic error of the lasso only depends on the link-function g via the parameters \mu and \sigma. Accounting for this and the fact that the result is *sharp*, shows that these two parameters fully capture the hardness of estimation (measured in squared-error) imposed by the link-function. In contrast in [19] the result is not sharp and the extra parameter \zeta in (7) is not necessary.

We thank the reviewer for the pointers to the references. Although dealing with a different algorithm, the paper by Yi et. al. is worth referencing. With regards to references on single-index models and the SDR literature please refer to the last paragraph of our response to Reviewer_6.

Revs_4&5
We thank both the reviewers for their positive feedback.

Rev_6
We thank the reviewer for the criticism. The literature on single index model/SDR is very long and studies the problem from a number of different perspectives and with different objectives. Thus, it would have been helpful if the reviewer had provided us with specific references that address the same question as our paper, and under the same setting (high-dimensions, asymptotics, regularized lasso, precise expressions).

To the best of our knowledge, when it comes to the single-index model for the measurements, our paper is the first to derive asymptotically precise results on the performance of any LASSO-type program. Moreover, some of our results are novel even under the prism of linear measurements. At this point, we highlight that even in the simpler setting of linear measurements, the precise error analysis of the lasso with Gaussian design had remained a challenging problem until only very recently ([9,18,22,26,27], lines 201-215). In our work, apart from keeping the Gaussian design assumption, we extend the model on the measurements, and also, we allow for more general regularizers and priors on x0 (cf. [9,26]).

We have identified the papers [5] and [19] as the ones that are most closely related to our setting, and have extensively discussed them in Sec 1.5.1. Due to space limitations, we have not been able to extensively review other papers from the area, ones that are not directly related to our result. We apologize in advance if we have missed any such references and we would greatly appreciate if the reviewer could point those to us.